EMBO
Molecular Medicine

# Infected erythrocytes and plasma proteomics reveal a specific protein signature of severe malaria

Jeremy Fraering[1,2], Virginie Salnot[2], Emilie-Fleur Gautier[2,3,4], Sem Ezinmegnon[5], Nicolas Argy[1,6],
Katell Peoc'h[4,7,8], Hana Manceau [ID][4,7,9], Jules Alao[10], François Guillonneau [ID][2,11,12],
Florence Migot-Nabias [ID][1], Gwladys I Bertin [ID][1,21✉], Claire Kamaliddin [ID][1,13,21✉] & NeuroCM consortium*

## Abstract

Cerebral malaria (CM), the most lethal complication of *Plasmodium falciparum* severe malaria (SM), remains fatal for 15–25% of affected children despite the availability of treatment. *P. falciparum* infects and multiplies in erythrocytes, contributing to anemia, parasite sequestration, and inflammation. An unbiased proteomic assessment of infected erythrocytes and plasma samples from 24 Beninese children was performed to study the complex mechanisms underlying CM. A significant down-regulation of proteins from the ubiquitin–proteasome pathway and an up-regulation of the erythroid precursor marker transferrin receptor protein 1 (*TFRC*) were associated with infected erythrocytes from CM patients. At the plasma level, the samples clustered according to clinical presentation. Significantly, increased levels of the 20S proteasome components were associated with SM. Targeted quantification assays confirmed these findings on a larger cohort ($n = 340$). These findings suggest that parasites causing CM preferentially infect reticulocytes or erythroblasts and alter their maturation. Importantly, the host plasma proteome serves as a specific signature of SM and presents a remarkable opportunity for developing innovative diagnostic and prognostic biomarkers.

**Keywords** Cerebral Malaria; LC-MS/MS; *Plasmodium falciparum*; Biomarkers; Red Blood Cells
**Subject Categories** Haematology; Microbiology, Virology & Host Pathogen Interaction; Proteomics

## Introduction

According to the World Health Organization's (WHO) report, there were 619,000 malaria-related deaths in 2022, and children under 5 years old were particularly at risk, representing 78.9% (488,391 cases) of fatal cases. Malaria is caused by the apicomplexan parasite *Plasmodium* spp, and infections with the species *Plasmodium falciparum* lead to the most severe cases. A significant proportion of malaria deaths results from cerebral malaria (CM) and, directly or indirectly, from anemia (WHO, 2022). The host–pathogen interactions define the dynamics between asymptomatic carriage, uncomplicated malaria (UM) and severe malaria (SM). SM is a complex pathology that leads to various specific clinical presentations and whose culmination is CM. It is estimated that 1–3% of cases of UM progress to severe disease; even when treated, severe malaria is fatal in 15–20% of cases. The WHO criteria for severe malaria definition in children are based on clinical assessment (such as consciousness, through the Blantyre score) and laboratory findings (including hemoglobin <5 g/dL and parasitemia) (Seydel et al, 2015; WHO, 2013).

Both the parasite genetics and the host's response may trigger different mechanisms of erythrocytes alterations, resulting in intra-vascular/extravascular hemolysis and/or impaired production of erythrocytes which contribute to the pathogenesis of severe malaria anemia (SMA). The pathophysiology of SM is multifactorial and remains partially understood. A better understanding of how the parasite adapts and impacts host physiology as the disease progresses is crucial for diagnosing and managing severe malaria.

One of malaria's most extensively studied pathophysiological mechanisms is the adhesion of infected erythrocytes (iEs) to the host's endothelial receptors (Wahlgren et al, 2017; Milner, 2018). This process, which is a parasite survival strategy to avoid the spleen clearance mechanism (Henry et al, 2020), is facilitated by variant surface antigens (VSAs) proteins, which are expressed by the parasite on the surface of the iE (Jensen et al, 2020; Kaestli et al, 2006; Lavstsen et al, 2012). Moreover, parasites causing CM present a higher proportion of circulating ring stage forms than in UM and asymptomatic individuals, suggesting a higher iE adherence through specific VSAs binding to the host endothelium, which could result in an accelerated increase of parasite burden in the host (Thomson-Luque et al, 2021; Andrade et al, 2020; Guillochon et al, 2022). Several intrinsic host factors, including endothelial dysfunction,

[1]UMR261 MERIT, Université Paris Cité, IRD, F-75006 Paris, France. [2]Plateforme Proteom'IC, Institut Cochin, Université Paris Cité, INSERM U-1016, CNRS UMR8104, Paris, France. [3]Institut Imagine-INSERM U1163, Hôpital Necker, Université Paris Cité, F-75015 Paris, France. [4]Laboratoire d'Excellence GR-Ex, F-75015 Paris, France. [5]Groupe de Recherche Action en Santé, Ouagadougou, Burkina Faso. [6]Laboratoire de parasitologie, Hôpital Bichat-Claude Bernard, APHP, Paris, France. [7]Biochimie Métabolique et Cellulaire, Hôpital Bichat-Claude Bernard, APHP, Paris, France. [8]Centre de Recherche sur l'Inflammation, UFR de Médecine Xavier Bichat, Université Paris Cité, INSERM UMR1149, Paris, France. [9]Département de Biochimie, Hôpital Universitaire Beaujon, APHP, Clichy, France. [10]Service de Pédiatrie, Centre Hospitalier Universitaire de la Mère et de l'Enfant-Lagune de Cotonou, Cotonou, Benin. [11]Unité OncoProtéomique, Institut de Cancérologie de l'Ouest, F-49055 Angers, France. [12]Université d'Angers, Inserm UMR 1307, CNRS UMR 6075, Nantes Université, CRCI2NA, F-49000 Angers, France. [13]Cumming School of Medicine, The University of Calgary, Calgary, AB, Canada. [21]These authors contributed equally: Gwladys I. Bertin, Claire Kamaliddin. *A list of authors and their affiliations appears at the end of the paper. ✉E-mail: gwladys.bertin@ird.fr; claire.kamaliddin@ucalgary.ca

inflammation, and coagulation, significantly also contribute to SM pathogenesis (Storm and Craig, 2014). An excessive and unregulated host immune response can lead to harmful effects on neighboring tissues causing the development of severe symptoms (Perkins et al, 2011). Infected erythrocytes adhering to host endothelial receptors trigger a cytokine-driven inflammatory response involving *IL1a*, *IL1b*, *IL6*, and *TNF*, intensifying fever and activating the innate immune response (Gillrie and Ho, 2016). This inflammatory response also upregulates adhesion molecules (*ICAM-1*, *VCAM-1*, and *E-selectin*) on endothelial cells, promoting iE binding to the vascular endothelium. These cell surface proteins are pivotal in the onset of severe clinical symptoms, such as anemia (Gillrie and Ho, 2016).

The parasite intra-erythrocytic development cycle leads to massive iE destruction after schizont rupture and is considered the most significant cause of anemia in infected patients. However, mechanisms such as uninfected erythrocyte (uE) lysis, dyserythropoiesis, iE and uE phagocytosis, iron restriction, and impaired erythropoietin (EPO) synthesis were also described to contribute to anemia in SM patients (Menendez et al, 2000). In addition, *P. falciparum* relies heavily on iron for crucial biological processes such as DNA replication and the synthesis of pyrimidines and heme, essential components for the parasite's multiplication (Clark et al, 2014; Sharma et al, 2021). *P. falciparum* is an auxotroph for iron, but the precise iron sources that the parasite can use are still not clearly defined. Hemoglobin, ferritin, and residual cytoplasmic iron are the main iron sources within the red blood cell, while in the extra-erythrocytic compartment, the transferrin protein is known to bind most serum iron (Clark et al, 2013).

Regarding host factors, Kumar et al found that platelet degranulation-associated proteins (*PLG, PF4, PNF-1*) were more abundant in the plasma of patients with SM than with UM (Kumar et al, 2020). More recently, an increased abundance of several circulating subunits of the circulating 20 S proteasome, along with downregulation of Insulin-like growth factor-1 (*IGF-1*), were observed in SMA Malian children plasma as compared with non-SMA anemic children (Mahamar et al, 2021). These findings raised the hypothesis that malarial anemia may be exacerbated by the hemolytic activity of the proteasome and the decreased erythropoiesis induced by *IGF-1* downregulation.

To further understand the host-pathogen interaction in pediatric severe malaria, we conducted an untargeted analysis of the plasma and iE from children living in seasonal malaria transmission areas and affected by *P. falciparum* malaria using cutting-edge mass spectrometry methods. By investigating both host and parasite proteins in iE and in the plasma, we present an integrative approach for studying severe malaria pathogenesis. Our study compared children affected by the uncomplicated form of the disease with those presenting the two major forms of severe malaria: cerebral malaria and severe malaria anemia. This study provides novel insights into the host–pathogen interactions contributing to severe forms of malaria.

# Results

## Enrollment and description of the cohorts

A total of 340 Beninese children samples were used in this study, including 103 patients of the primary study cohort called CIVIC (49 CM, 33 UM, and 21 SMA) (Kamaliddin et al, 2019); 170 patients of the validation cohort NeuroCM (74 CM and 96 UM) and 67 non-infected (NI) control patients. Out of the 340 samples collected, 314 were available in sufficient quantity to conduct colorimetric analyses. No additional inclusion criteria were applied. The mean age of the included patients from the CIVIC cohort was 28.94 [s.d. = 15.60] months, 43% of the patients were female, and the mean weight was 10.82 kg [s.d. = 2.96] (Table 1). No significant differences for age ($p = 0.1004$), sex ($p = 0.541$), and weight ($p = 0.4914$) were observed between the three clinical groups (UM, CM, and SMA; Table 1). In the severe malaria patients (CM and SMA), the mean hemoglobin concentration was significantly ($p = 0.0015$) lower in SMA compared to CM samples (4.27 g/dL and 5.61 g/dL, respectively). Among the severe malaria patients, 27.5% deceased in the 30 days after inclusion.

Figure 1 presents a concise summary of the workflow employed in this study for iE and plasma proteomics. Briefly, for iE proteome analysis, 17 samples were selected from the CIVIC divided into three CM, eight SMA, and six UM samples. The selection was based on successful maturation and MACS enrichment, and these samples were subjected to LC-MS/MS analysis. For the plasma proteome, 24 samples were selected, encompassing the previously mentioned iE samples for a total of six CM, six SMA, and six UM cases, along with six non-infected (NI) samples. The clinical data (Table 1; mean [min−max]; *p* value all cohort vs. sub-set) from this sample subset was representative of the whole cohort in terms of age (35.89 [7–60] months; $p = 0.122$), weight (10.60 [6.3–15.4] kg; $p = 0.770$), and sex ratio (41% female) and presented a non-significant lower mean parasitemia (174,765 parasites/µL) when compared to the whole CIVIC cohort (546,447 parasites/µL; $p = 0.809$).

## Descriptive analysis of iE and plasma samples by LC-MS/MS showed specific clustering between the clinical groups

Regarding the iE, a total of 1417 human proteins (1424 were identified—Fig. 2A left) and 1846 parasitic proteins (1859 identified —Fig. 2A middle) were quantified by LC-MS/MS across all patients (Dataset EV1). Among the 1417 human proteins, 426 were exclusively found in SMA presentations and are mainly involved in translation initiation, protein refolding and sugar catabolism pathways. Regarding the parasitic proteins, 680 were exclusive of SMA and were mainly involved in rRNA processing and mRNA splicing via spliceosome pathways (Dataset EV2). For the plasma a total of 2422 proteins were identified (Dataset EV3) and 1434 were quantified. Among the 1434 plasma proteins, 114 were only quantified in the CM group. These exclusive proteins were related to DNA replication, intra-Golgi vesicle-mediated transport, mRNA splicing through spliceosome, and DNA repair pathways (Dataset EV4; Appendix Fig. S1). Nevertheless, there were less than ten specific proteins for the other clinical groups: UM, SMA, and NI (Fig. 2A right).

A principal component analysis (PCA) on plasma samples revealed three distinct groups; NI, CM + SMA, and UM (Fig. 2B). NI samples clustered at the right side of the PCA distinctly from infected patients (UM, SMA, and CM) through the first component. The CM and SMA samples clustered together through the first component (Fig. 2B). UM samples appeared

**Table 1. Clinical data for the CIVIC patients enrolled.**

| | CIVIC cohort | | | | | CIVIC sub-set MS/MS | | | | | |
| --- | --- | --- | --- | --- | --- | --- | --- | --- | --- | --- | --- |
| | All patients (n = 103) | CM (n = 49) | UM (n = 33) | SMA (n = 21) | p value | All sub-set MS/MS (n = 24) | CM (n = 6) | UM (n = 6) | SMA (n = 6) | NI (n = 6) | p value |
| Age (month) mean (s.d.) | 28.94 (15.60) | 31.69 (16.54) | 23.38 (14.08) | 28.86 (13.84) | 0.1004 | 35.89 (19.72) | 39.83 (19.72) | 42.17 (21.80) | 25.67 (16.31) | NA | 0.3094 |
| Female gender, n (%F) | 42 (43%) | 22 (46%) | 14 (45%) | 6 (32%) | 0.5413 | 7 (41%) | 2 (33%) | 4 (66%) | 1 (17%) | NA | NA |
| Weight (kg) mean (s.d.) | 10.82 (2.96) | 11.02 (3.05) | NA | 10.33 (2.73) | 0.4914 | 10.60 (3.63) | 11.65 (3.85) | NA | 9.76 (3.64) | NA | 0.6746 |
| Hemoglobin (g/dL) mean (s.d.) | 5.20 (1.72) | 5.61 (1.84) | NA | 4.27 (0.86) | 0.0015 | 4.51 (0.84) | 4.40 (1.14) | NA | 4.62 (0.48) | NA | 0.9654 |
| Parasite_density (parasite/µL) (s.d.) | 546,447 (932,942) | 967,978 (1,126,030) | 46,804 (165,746) | 133,897 (242,892) | <0.0001 | 174,765 (247,787) | 222,219 (214,449) | 33,231 (45,593) | 268,844 (3,511,778) | NA | 0.2279 |
| Blantyre score median [Q1-Q3] | 2 [2-4] | 2 [1-2] | NA | 5 [4-5] | <0.0001 | 3 [2-5] | 2 [1-2] | NA | 5 [4-5] | NA | NA |
| Death (% death) | 19 (27.5%) | 18 (37.5%) | NA | 1 (4.77%) | NA | 1 (8.5%) | 1 (17%) | NA | 0 (0%) | NA | NA |

Multiple comparisons were made using ANOVA tests (cut-off p value set to 0.05), and two-group comparisons were made using Mann–Whitney U-test. Categorical values (sex ratio) were compared using Chi-square test. No clinical data were collected from the non-infected samples (NI). Significant p values are presented in bold.
UM uncomplicated malaria, SMA severe malarial anemia, NA not applicable.

predominantly distinct but closer to NI (except UM3 and UM5) than CM and SMA.

A differential abundance analysis of plasma proteins showed 33 differentially abundant proteins (DAP) (q < 0.05), between NI and the infected samples. Among these DAP, the levels of the human proteins CRP (21.70 [+/−1.59] Log2LFQ), Hemoglobin A1 (HBA1; 21.97 [+/−1.38] Log2LFQ), Hemoglobin B (HBB; 22.07 [+/−1.46] Log2LFQ) and Myeloperoxidase (MPO; 15.40 [+/−1.63] Log2LFQ) were higher in infected samples than in NI (15.09 [+/−4.18] Log2LFQ with FC = 98; 18.08 [+/−2.58] Log2LFQ with FC = 15; 18.21 [+/−2.45] Log2LFQ with FC = 15 and 12.03 [+/−1.82] Log2LFQ with FC = 10, respectively). Moreover, these protein abundances highly influence the sample localization through the first axe of the general PCA (Fig. 2B,C). Altogether, the PCA analysis on the global plasma proteome suggests that patients with severe malaria displayed a specific protein signature compared to UM and NI.

### The iE host proteome analysis revealed an increased transferrin receptor 1 (TFRC) abundance and a dysregulated ubiquitination pathway in cerebral malaria patients

To better characterize the human proteome variations associated with severe malaria, a differential abundance analysis was performed between the clinical groups. In infected erythrocytes, 14/579 differentially abundant proteins were identified between CM and UM samples. Notably, five proteins (CUL2, YOD1, PSMD2, FBXO7, and UBAC1) implicated in protein ubiquitination and degradation through proteasome pathways were less abundant in CM samples (17.24 [+/−0.44]; 20.14 [+/−0.12]; 20.86 [+/−0.65]; 20.42 [+/−0.21] and 18.77 [+/−1.02] Log2LFQ) compared to UM samples (18.46 [+/−0.43] Log2LFQ with p = 0.0147; 20.94 [+/−0.44] Log2LFQ with p = 0.0250; 21.64 [+/−0.33] Log2LFQ with p = 0.0443; 21.76 [+/−0.47] Log2LFQ with p = 0.0026 and 20.47 [+/−0.86] Log2LFQ with p = 0.0325) (Fig. 3A; Dataset EV5). F-box only protein 7 (FBXO7) and Ubiquitin thioesterase (YOD1) were also decreased in SMA compared to UM samples. A significant increase of Cathepsin G (CTSG) abundance in CM (23.60 [+/−1.63] Log2LFQ) when compared to UM (19.46 [+/−1.98] Log2LFQ with p = 0.023) was also identified. Interestingly, the levels of Alpha-Actinin-4 (ACTN4) and Transferrin receptor protein 1 (TFRC) were found significantly increased in CM (21.24 [+/−0.76] Log2LFQ; 22.33 [+/−0.60] Log2LFQ respectively) compared to UM samples (16.91 [+/−0.74] Log2LFQ with p = 0.0025 and FC = 20; 18.41 [+/−1.46] Log2LFQ with p = 0.0124 and FC = 15.2), but not between SMA (17.73 [+/−1.84] Log2LFQ with p = 0.424; 18.91 [+/−1.99] Log2LFQ with p = 0.712) and UM samples suggesting a specific CM phenotype (Fig. 3B).

Moreover, we observed by targeted ELISA assay that TFRC abundance was higher in CM patients (31.70 [+/−14.34] ng/mL; n = 10) than in UM (23.03 [+/−14.06] ng/mL; n = 10) (Figure EV1) but the difference was not significant (p = 0.188).

In addition, a significant over-representation of iron homeostasis pathway (p < 0.0001) and Ferroptosis signaling pathways (p = 0.0116) was identified in CM compared to SMA using an over-representation analysis (ORA) with Ingenuity Pathway Analysis (IPA) (Appendix Fig. S2). Furthermore, protein ubiquitination

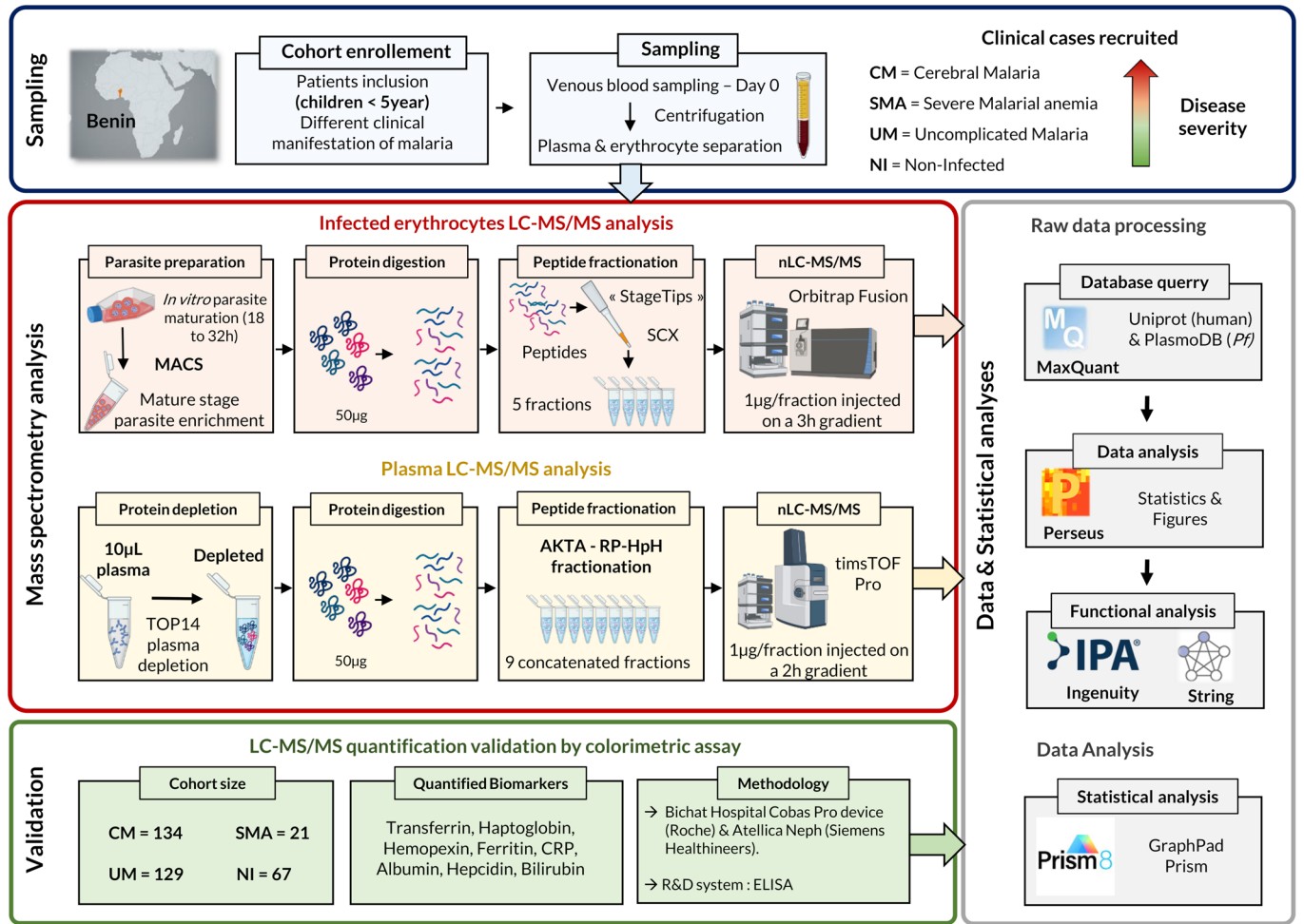

**Figure 1.  Study workflow.**

Schematic representation of the methodological approach used to quantify by LC-MS/MS protein abundances in infected erythrocytes and plasma samples from Beninese children suffering from different clinical manifestations of malaria. Only a single technical replicate was conducted, and it was performed using only biological replicates for both LC-MS/MS and colorimetric results (MACS magnetic activated cell sorting, SCX strong cation exchange, RP-HpH reverse-phase high pH chromatography, nLC-MS/ MS nano-liquid chromatography tandem mass spectrometry).

pathway, *FAT10*, and *BAG2* signaling pathways were also significantly ($p = 0.011$; $p = 0.033$, and $p = 0.048$, respectively) over-represented in CM and SMA compared to UM samples (Appendix Fig. S2).

## Host metabolism, inflammatory response, and circulating 20S proteasome are associated with malaria severity

Next, we investigated the global response to *P. falciparum* infection by analyzing the plasmatic proteome and found 68 DAP ($q$-value $< 0.05$) among the 781 selected proteins (Dataset EV6). Of these 68 DAP, 75% ($n = 54/68$) presented significantly different abundances in CM *vs.* UM (Fig. 4A). More specifically, in the CM samples, *MPO* was more abundant ($q = 0.0004$; FC = 5.31). On contrary, three complement components (*CFP, C3* and *C1R*) and three coagulation factors (*F13A1, F12* and *PF4*) were less abundant in CM (17.28 [+/−1.02]; 22.67 [+/−0.19]; 20.31 [+/−0.64]; 16.83 [+/−0.68]; 18.38 [+/−0.99]; 16.92 [+/−0.61] Log2LFQ

respectively) when compared to UM samples (18.87 [+/−0.49] Log2LFQ with $q = 0.0262$; 23.19 [+/−0.37] Log2LFQ with $q = 0.0153$; 21.07 [+/−0.28] Log2LFQ with $q = 0.0329$; 18.28 [+/−0.84] Log2LFQ with q = 0.0098; 20.24 [+/−0.45] Log2LFQ with $q = 0.0080$; 18.75 [+/−0.85] Log2LFQ with $q = 0.0004$, respectively). In addition, *F12* was also decreased in CM samples when compared to SMA (FC = −2.35). A significant increase of Vascular cell adhesion molecule 1 (*VCAM1*), Cathepsin D, *HSP90AA1*, and *HSPA4* (FC = 1.9; 1.7; 1.0 and 1.5, respectively, with a $q < 0.05$) was observed in CM samples when compared to UM and SMA samples. A significant ($q = 0.0165$) decrease of the heme scavenger protein Hemopexin (*HPX*) abundance was identified in CM (20.79 [+/−1.51] Log2LFQ) compared to UM (22.93 [+/−0.68] Log2LFQ with $q = 0.0165$ and FC = −4.426) and SMA (22.35 [+/−0.60] Log2LFQ with $q = 0.0165$ and FC = −2.961) (Fig. 4A; Dataset EV6).

Moreover, a significantly higher abundance ($q < 0.05$) of seven subunits of the circulating 20S proteasome (*PSMA1/4/5/6/7*, and

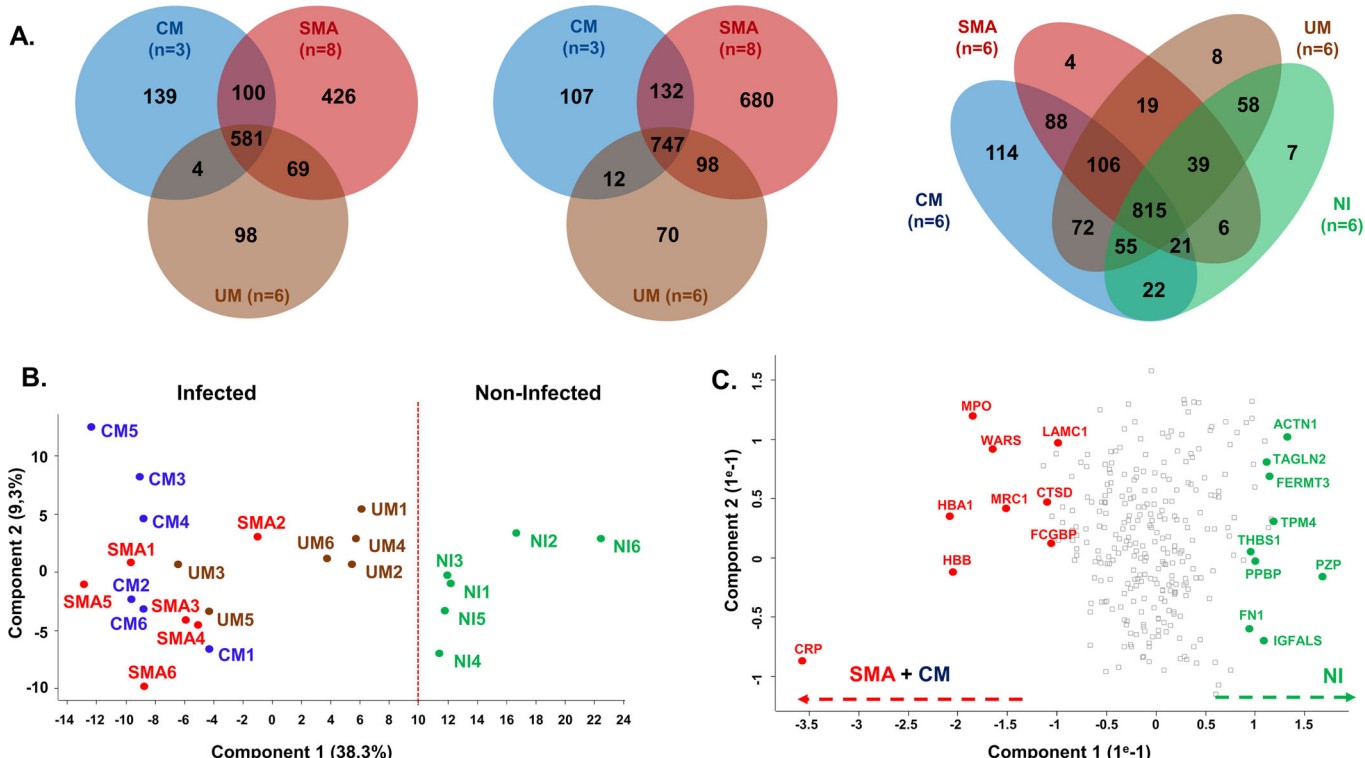

**Figure 2. Global LC-MS/MS descriptive analysis and principal component analyses of plasma samples.**

(A) Venn diagrams of the human (Left) and parasitic (Middle) proteins found in iE and in plasma (Right) from pediatric patients suffering from cerebral malaria (CM), severe malaria anemia (SMA), uncomplicated malaria (UM) and non-infected patient (NI). Diagrams were generated using jvenn online (81) with a distinct color code for the different malaria clinical groups; CM in Blue, SMA in Red, UM in Brown, and NI in Green. (B) Principal Component Analysis (PCA) on proteins quantified in all 24 plasma samples ($n = 284$ proteins), generated with the log2 of LFQ values calculated by MaxQuant algorithm. Color code: CM in Blue, SMA in Red, UM in Brown, and NI in Green. The red dotted line discriminates NI samples from Infected samples. (C) Protein-based PCA ($n = 284$), in red proteins with an increased impact on sample clustering in the left side of the PCA in (B), and in green proteins with a high impact on sample clustering in the right side of the PCA in (B).

*PSMB1/5*) was identified in CM samples, when compared to UM samples. Four of these 20S proteasome subunits (*PSMA4/6/7* and *PSMB5*) were also increased in SMA *vs.* UM (Fig. 4A; Dataset EV6). Furthermore, a PCA of the 15 distinct sub-units of the 20S proteasome (Fig. 4B; Dataset EV7) showed a clustering of UM and NI samples together, while CM and SMA were located in a separate cluster.

The IPA analysis on plasma revealed that the ubiquitin-like *FAT10*, the co-chaperone *BAG2* and the protein ubiquitination pathways were significantly enriched in UM *vs.* CM and UM vs. SMA (Appendix Fig. S3). To summarize, an increased abundance of inflammation markers and a decreased abundance of coagulation and complement components were measured in samples from CM children. In addition, the quantification from 15 sub-units of the 20S proteasome seems to perfectly separate the SM group (CM + SMA) from UM and NI.

## LC-MS/MS analysis of iE and plasma parasitic proteins suggests a deregulated growth rate and a more efficient host invasion mechanism in SM isolates

By LC-MS/MS, 619 *P. falciparum* proteins were quantified in iE samples (Dataset EV8). A significant upregulation of two chaperone proteins implicated in protein refolding: DnaJ protein putative—*PF3D7_0823800* and Putative knob associated

heat shock protein 40—*PF3D7_0201800* was identified in CM isolates (21.12 [+/−0.41]; 20.04 [+/−0.07] Log2LFQ) compared to UM (18.56 [+/−1.02] Log2LFQ with $p = 0.022$; 18.31 [+/−0.81] Log2LFQ with $p = 0.035$, respectively). Most parasitic DAP were related to metabolic pathways such as the polyamine biosynthetic process, the organic acid metabolic process, DNA replication, and protein refolding system (Appendix Fig. S4). Moreover, serine/threonine protein phosphatase *UIS2* (PF3D7_1464600) was found to be significantly upregulated in SMA (19.31 [+/−0.57] Log2LFQ) compared to UM samples (18.08 [+/−0.66] Log2LFQ with a $p = 0.031$ and FC = 2.35). Additionally, a significant over-representation ($p = 0.0082$) of proteins from the minichromosome maintenance complex (*MCM*), which controls DNA replication, was found in CM vs. SMA.

In the plasma, 119 parasitic proteins were identified, but only 29 could be quantified. Among the latter 29 proteins, 13 were found in all infected groups, and more interestingly 11 proteins were only quantified in SM. Four of these latter proteins are surface proteins (*MSP9*, *MSP6*, antigen 332, 6-cystein P92) expressed by the parasite on the surface of the iE and involved in erythrocyte invasion (Dataset EV9). No statistically significant difference in protein abundance was found between the clinical groups. Altogether, the results on parasitic proteins

 

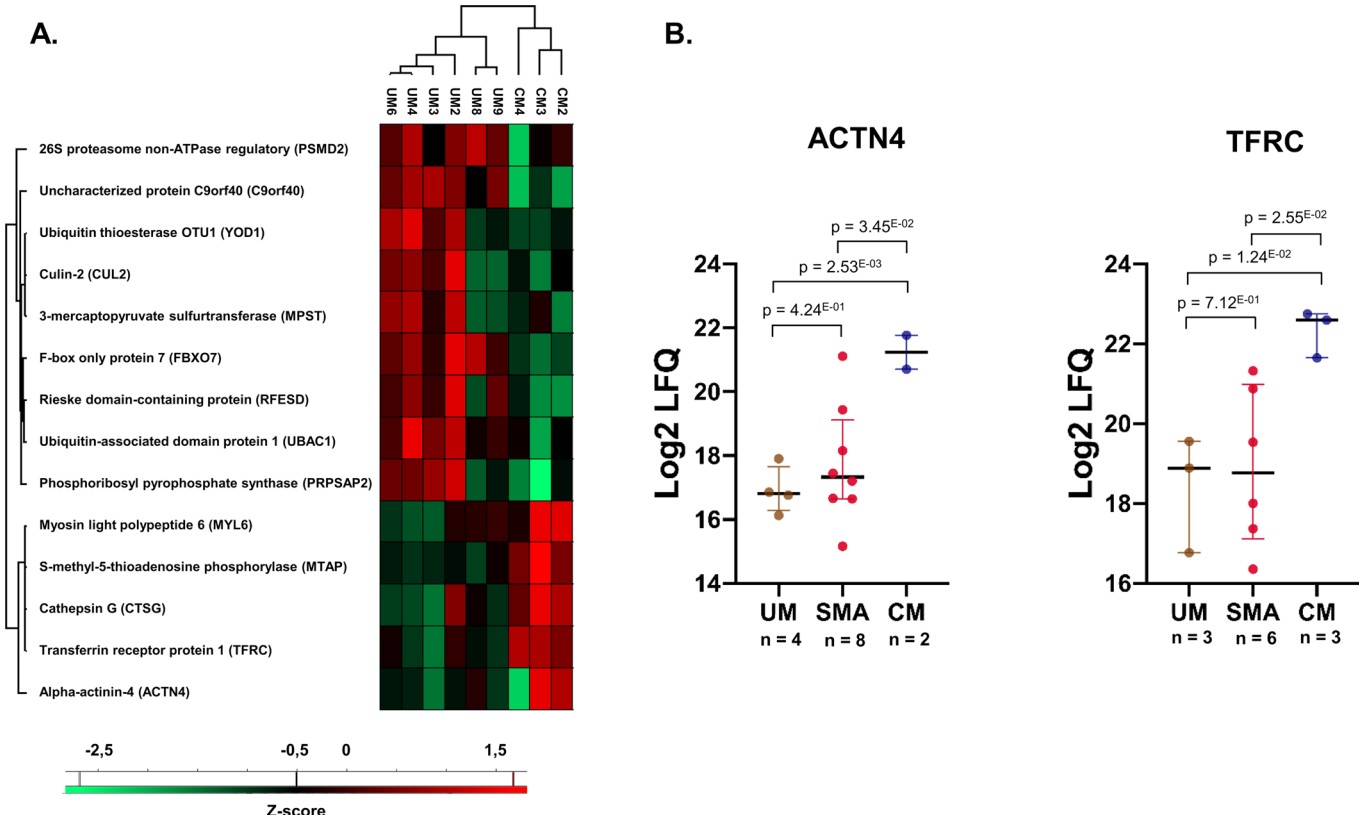

**Figure 3. Differentially abundant host proteins in iE samples analyzed by LC-MS/MS.**

(A) Heatmap representing Z-scored Log2LFQ values of the 14 differentially abundant proteins identified by LC-MS/MS analysis in between CM ($n = 3$) and UM ($n = 6$) biological replicates of iE samples ($p < 0.05$; Student's $t$ test). Heatmaps with Euclidean distances between clinical groups were generated using Perseus software (v1.6.15). Log2LFQ missing values were imputed following a normal distribution (width = 0.3 and 1.8 down shift) to generate the heatmap. (B) Dot plot (GraphPad Prism 8.0) representation with median (black bold line) and colored bars (Q1–Q3) of Log2 LFQ values for *ACTN4* and *TFRC* between the three clinical groups showed a significant increase of *TFRC* and *ACTN4* abundance exclusively in CM samples. Student's $t$ test $p$ values are displayed in the figure. Biological replicate numbers for each group are depicted in the figure. Source data are available online for this figure.

in CM showed an over-representation of proteins involved in different biosynthesis process, DNA replication, and erythrocyte invasion.

## Validation of LC-MS/MS data and assessment of iron metabolism and inflammation pathways by selective quantification of nine human plasma proteins

Levels of nine plasma proteins associated with iron metabolism and inflammation were assessed in 340 plasma samples (Dataset EV10). All results will be presented as follows: "clinical group" (mean [CI$_{95\%}$low-CI$_{95\%}$up]; Mann–Whitney $U$-test $p$ value).

The *CRP*, *HPX* and Ferritin levels found by targeted quantification between each clinical group were similar to the LC-MS/MS analysis (Fig. 5; Dataset EV10). However, contrary to the observations from the LC-MS/MS analysis of *HPX*, the colorimetric results showed no significant differences between SMA and CM samples.

A significant increase of *CRP* in CM (164.8 [148.80–180.70] mg/L; $p = 1.42^{E-11}$) and SMA (148.3 [93.88–202.70] mg/L; $p = 1.76^{E-02}$) when compared to UM (88.76 [76.82–100.70] mg/L) and a significant

decrease of Albumin in CM (26.52 [25.17–27.86] g/L; $p = 1.82^{E-10}$) and SMA (27.07 [25.03–29.10] g/L; $p = 3.37^{E-04}$) compared to UM (32.82 [31.72–33.93] g/L) were found (Fig. 5). Regarding erythrocyte lysis markers, Haptoglobin was less abundant in CM (0.1286 [0.0976–0.1596] g/L; $p = 7.95^{E-10}$) and SMA (0.1029 [0.0967–0.1090] g/L; $p = 9.25^{E-03}$) compared to UM (0.4320 [0.3325–0.5316] g/L) and Hemopexin was also less abundant in CM (0.3041 [0.2675–0.3407] g/L; $p = 1.50^{E-27}$) and SMA (0.2991 [0.2174–0.3808] g/L; $p = 7.77^{E-08}$) compared to UM (0.7442 [0.6978–0.7906] g/L) samples. Additionally, total bilirubin concentration was higher in CM (21.76 [17.62–25.90] μmol/L; $p = 2.71^{E-19}$) than in UM (6.059 [5.304–6.813] μmol/L) and SMA (7.365 [3.734–11.000] μmol/L; $p = 2.97^{E-04}$). Conjugated bilirubin was also increased in CM (10.24 [8.101–12.37] μmol/L; $p = 4.01^{E-16}$) when compared with UM (3.202 [3.046–3.358] μmol/L) and SMA (4.077 [2.483–5.670] μmol/L; $p = 3.52^{E-03}$) (Fig. 5).

Regarding iron metabolism biomarkers, an increase in Hepcidin concentrations was associated with malaria infection compared to NI samples. However, no significant differences were found between the three infected groups (Fig. 5). Moreover, plasma transferrin concentration was decreased in CM (1.546 [1.449–1.644] g/L; $p = 5.10^{E-07}$) samples when compared to

 

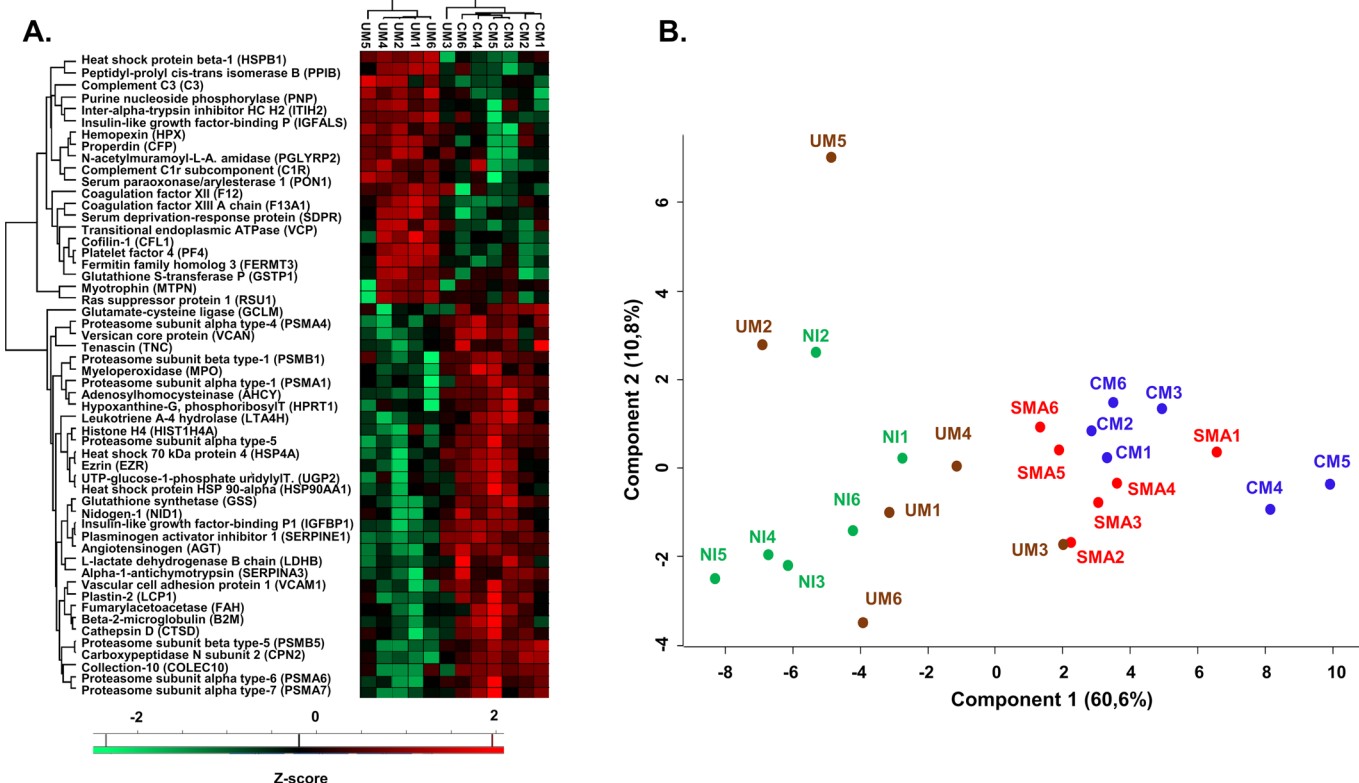

**Figure 4.    Assessment of Severe Malaria protein signature in plasma through LC-MS/MS analysis.**

(A) Heatmap representing $Z$-scored log2LFQ values of the 54 differentially abundant proteins in CM ($n = 6$) and UM ($n = 6$) biological replicates of plasma samples (ANOVA with Benjamini–Hochberg correction for multiple test: $q$-value < 0.05, and Tukey's HSD test). LFQ missing values were imputed with normal distribution values (width = 0.3 and 1.8 down shift) to establish the heatmap (Euclidean distances). (B) PCA analysis of the 15 circulating 20S proteasome sub-units quantified in plasma samples. Biological replicates: CM ($n = 6$) in Blue, SMA ($n = 6$) in Red, UM ($n = 6$) in Brown, and NI ($n = 6$) in Green.

UM (1.873 [1.800–1.945] g/L) and SMA (1.888 [1.591–2.184] g/L; $p = 3.35^{E-02}$).

## Discussion

The pathophysiology of cerebral malaria (CM) remains poorly understood. To perform an unbiased study of CM, a non-targeted proteomic approach was implemented on a sample set of 17 iE and 24 plasma samples from children with different malaria clinical presentations (NI, UM, SMA, and CM) (Fig. 1). To the best of our knowledge, the dataset generated in this study represents one of the most robust datasets ever obtained using this approach on such samples.

In infected erythrocytes from CM patients, a reduction in the expression of five host proteins playing a critical role in intracellular protein degradation by the proteasome was identified (Fig. 3A). In addition, significantly higher levels of transferrin receptor protein 1 (*TFRC*) and alpha-actinin 4 (*ACTN4*) proteins were identified in CM isolates (Fig. 3B). The tendency of increased TFRC levels in CM iE samples was also found through targeted quantification of *TFRC* using independent samples from the NeuroCM cohort (n = 20) but was not significant ($p = 0.19$). This result suggests CM-specific regulatory mechanisms, either related

to TFRC recycling or erythrocyte maturation processes. *ACTN4*, a protein linked to actin, aids *TFRC* recycling via the CART complex interaction (Yan et al, 2005), and increased levels of *ACTN4* could upregulate *TFRC* internalization. Interestingly, *TFRC* is degraded during reticulocyte maturation into erythrocyte (Moura et al, 2015; Gautier et al, 2018; Sae-Lee et al, 2022). Moreover, considering the absence of a nucleus and mRNA within mature erythrocytes, and the subsequent absence of further protein synthesis, the presence of the *TFRC* must have preceded the parasite's infection of the red blood cell. A hypothesis to explain these findings is that the parasite causing CM preferably infects reticulocytes and disrupts their maturation process into mature erythrocytes, possibly by maintaining the *TFRC* for its own iron requirements. *P. falciparum* is already known to infect reticulocytes (Pasvol et al, 1980), and higher reticulocyte counts have been reported in children presenting with SMA when compared with UM in Kenya (Novelli et al, 2010), indicating a greater availability of circulating reticulocytes in SM cases. The mechanisms that could lead to a higher reticulocytes infection in CM patients might be linked to specific protein-protein interactions similar to what is observed for *P. vivax*, which binds *TFRC* via the *RBP2b* protein to target reticulocyte cells specifically (Gruszczyk et al, 2018). Otherwise, by parasitizing erythroid precursors in the bone marrow (Neveu et al, 2020; Aguilar et al, 2014; Smalley et al, 1981), *P. falciparum* would probably experience

 

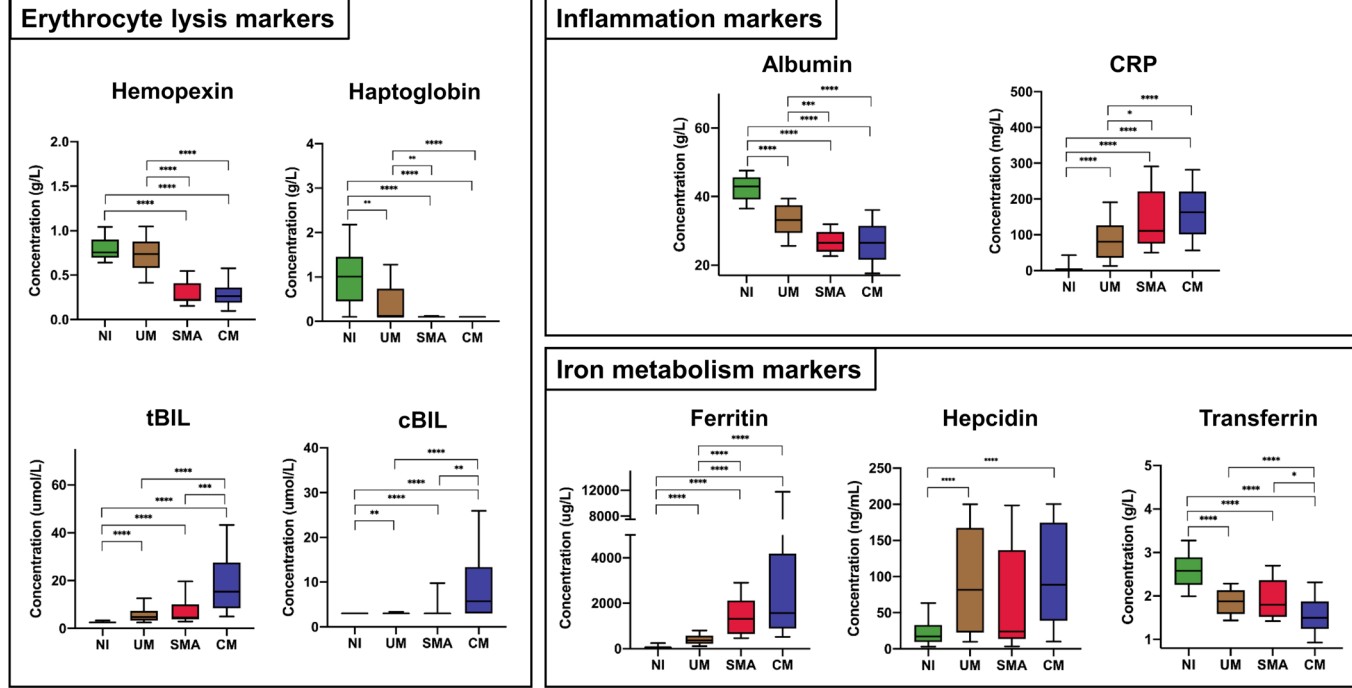

**Figure 5.  LC-MS/MS quantification validation by colorimetric dosage of nine proteins in plasma.**

Quantification of Albumin, CRP, total and conjugated Bilirubin, Ferritin, Hemopexin, Hepcidin, Transferrin, and Haptoglobin in the plasma of Beninese children according to the clinical presentation of malaria ($n = 340$ biological replicates). Data were analyzed with GraphPad Prism 8.0. Bars correspond to 10–90 percentiles, with a box composed of [Q1; median; Q3]. Mann–Whitney $U$-test results are represented in the figure (*$p$ value < 0,05); **$p$ value (<0,01); ***$p$ value (<0,001), and ****$p$ value (<0,0001)). All Mann–Whitney exact $p$ values will be presented below as follows: Biomarker (NI vs. UM; NI vs. SMA; NI vs.CM; UM vs. SMA; UM vs. CM and CM vs. SMA). Hemopexin (1.00$^{E\text{-}1}$; 6.72E$^{\text{-}09}$; 1.97E$^{\text{-}24}$, 7.77E$^{\text{-}08}$; 1.50E$^{\text{-}27}$ and 9.97E$^{\text{-}01}$), Haptoglobin (2.11E$^{\text{-}07}$; 7.28E$^{\text{-}06}$; 9.19E$^{\text{-}21}$; 9.25E$^{\text{-}03}$; 7.95E$^{\text{-}10}$ and 9.77E$^{\text{-}01}$), tBIL (1.88E$^{\text{-}13}$; 5.28E$^{\text{-}07}$; 2.48E$^{\text{-}19}$; 5.90E$^{\text{-}01}$; 2.71E$^{\text{-}19}$ and 2.97E$^{\text{-}04}$), cBIL (2.77E$^{\text{-}02}$; 2.76E$^{\text{-}03}$; 9.99E$^{\text{-}11}$; 2.09E$^{\text{-}01}$; 4.01E$^{\text{-}16}$ and 3.52E$^{\text{-}03}$), Albumin (2.40E$^{\text{-}14}$; 1.86E$^{\text{-}07}$; 1.85E$^{\text{-}19}$; 3.37E$^{\text{-}04}$; 1.82E$^{\text{-}10}$ and 8.18E$^{\text{-}01}$), CRP (7.42E$^{\text{-}14}$; 1.11E$^{\text{-}09}$; 1.77E$^{\text{-}16}$; 1.76E$^{\text{-}02}$; 1.42E$^{\text{-}11}$ and 4.64E$^{\text{-}01}$), Ferritin (5.16E$^{\text{-}13}$; 1.74E$^{\text{-}08}$; 1.00E$^{\text{-}17}$; 1.60E$^{\text{-}05}$; 1.42E$^{\text{-}24}$ and 2.10E$^{\text{-}01}$), Hepcidin (2.95E$^{\text{-}10}$; 9.95E$^{\text{-}2}$; 2.63E$^{\text{-}13}$; 1.39E$^{\text{-}01}$; 3.05E$^{\text{-}01}$ and 6.49E$^{\text{-}02}$), and Transferrin (2.24E$^{\text{-}13}$; 2.52E$^{\text{-}04}$; 4.44E$^{\text{-}16}$; 6.80E$^{\text{-}01}$; 5.10E$^{\text{-}07}$ and 3.35E$^{\text{-}02}$). Source data are available online for this figure.

a faster proliferation, which could result in a substantial and exponential increase in parasite load. Furthermore, the reticulocyte's metabolome is more diverse and enriched than mature erythrocytes, giving a potential advantage in nutrient availability for the growth of *P. falciparum* (Srivastava et al, 2015, 2017). Altogether, these results support the hypothesis of recent report studies indicating a shorter circulation time of *P. falciparum* in CM patients (Andrade et al, 2020; Guillochon et al, 2022; Tonkin-Hill et al, 2018). This decreased circulation time, linked to parasite sequestration, leads to increased parasitemia in CM patients. However, the results obtained from infected erythrocytes might be prone to a high false discovery rate due to a small sample size and the statistical test employed.

In addition to the iE proteome, we studied the plasma proteome of children in the CIVIC cohort. The clustering of the identified proteins highlighted specific protein profiles associated with severe malaria, supporting the hypothesis of a distinct plasma protein signature in CM and SMA. Interestingly, the abundance of several subunits of the circulating 20 S proteasome was higher in samples from patients with CM or SMA when compared to UM samples. Furthermore, a PCA of the fifteen proteasome sub-unit concentrations (Fig. 4B) demonstrated a clear separation between SM (CM and SMA), and UM and NI. These results suggest that circulating proteasome levels could serve as a biomarker for disease severity.

Importantly, these results are in agreement with a recent mass spectrometry analysis of plasma from SMA children (Mahamar et al, 2021). In the latter study, the authors suggested that increased circulating 20S proteasome levels worsened malarial anemia through increased hemolysis and dyserythropoiesis. Functional analysis of the deregulated proteins between CM or SMA, when compared to UM showed the following over-represented pathways: HLA-F adjacent transcript 10 (*FAT10*) and Bcl-2-associated athanogene 2 (*BAG2*) signaling, and protein ubiquitination. *FAT10* has a structure close to ubiquitin, implying a shared function in protein degradation via the 26S proteasome (Hipp et al, 2005). *BAG2*, linked to *HSP70*, aids misfolded protein refolding and proteolysis (Rauch and Gestwicki, 2014). These pathways were also identified in iE dysregulated proteins, suggesting stress-induced protein degradation or refolding in severe malaria. Moreover, complement components and coagulation factors abundances were lower in CM compared to UM and SMA samples. Both latter pathways were also over-represented in the functional analysis, suggesting a potential impairment of parasite clearance and endothelial repair system, as previously reported (Kumar et al, 2020). Aside from protein degradation and proteasome pathways, we investigated proteins involved in anemia. Haptoglobin and Hemopexin were less abundant in CM than in UM plasma samples, as previously reported (Kumar et al, 2020). Conversely, bilirubin

concentrations were significantly higher in CM than in UM and SMA plasma samples (Fig. 5). Together, these results show that hemolysis is more important in SM samples, as expected, and that the type of hemolytic anemia in CM presentations compared with SMA could be different. Indeed, Hemopexin is known to play a critical role in heme transport and degradation and is a marker of hemolysis severity (Delanghe and Langlois, 2001). In addition, previous studies have demonstrated that the internalization of the Heme-Hemopexin complex in hepatocytes leads to the induction of ferritin synthesis (Hentze and Kühn, 1996; Klausner et al, 1993). This is consistent with the increased ferritin abundance observed in CM plasma samples (Fig. 5). Ferritin is a protein involved in iron storage, primarily in the liver and macrophages. It plays a critical role in iron metabolism, and elevated ferritin concentration in plasma is a marker of iron overload or inflammation (Wang et al, 2010) and was correlated with malaria severity in children from Papua New Guinea (O'Donnell et al, 2009). Additionally, several studies have suggested that ferritin plays a role in immunity (Wang et al, 2010; Hazard and Drysdale, 1977; Matzner et al, 1979) by decreasing hematopoiesis, and particularly the biosynthesis of macrophages and erythroid precursors (Broxmeyer et al, 1986, 1989). The lower concentration of transferrin (*TF*) in the plasma of CM samples appeared to be consistent with the higher abundance of *TFRC* in the iEs from the CM group. These findings strengthen the hypothesis that parasites use host *TFRC* for their own iron needs by binding host transferrin from plasma.

Anemia of inflammation is primarily caused by functional iron deficiency and reduced erythropoiesis caused by systemic inflammation triggering immune cell activation and cytokine production (Weiss et al, 2019). Hepcidin is the main hormone implicated in iron homeostasis, and increases during inflammation (Spottiswoode et al, 2014). Hepcidin also blocks iron within macrophages and hepatocytes through ferroportin endocytosis (Nemeth et al, 2004). Hepcidin concentration was approximately threefold higher in CM and UM plasma samples, when compared to NI, indicating a host response to the infection by *P. falciparum*. Consequently, hepcidin will decrease iron availability for erythropoiesis and could contribute to the dyserythropoiesis observed in *P. falciparum* malaria (Weiss et al, 2019; Ganz and Nemeth, 2012; Aschemeyer et al, 2018).

The plasma compartment contains *P. falciparum* proteins that are either secreted directly by the parasites or released during the merozoites egress. This study identified 119 *P. falciparum* proteins, and 29 were quantified. Among the 29 quantified parasitic proteins, we identified *HSP70* (PF3D7_0917900), involved in protein refolding and the glycolysis protein enolase (PF3D7_1015900). Parasitic chaperone proteins like *HSP70* are parasite survival factors, counteracting several biological host reactions including fever, degradation or immune system response (Barth et al, 2022), and contribute to *P. falciparum* virulence (Shonhai and Blatch, 2021). The glyceraldehyde-3-phosphate dehydrogenase (*GAPDH*; PF3D7_1462800) was quantified only in the SM groups (Dataset EV9), representing an indirect reflection of increased protein levels. *GAPDH* is involved in glucose metabolism and host cell entry (Bahl et al, 2003), and was recently proposed as a biomarker for malaria (Krause et al, 2017). Among the 119 *P. falciparum* proteins found in plasma, 11 were identified exclusively in SM samples (Dataset EV9), and could serve as potential novel biomarkers for early detection of severe malaria development, as

they were not detected in UM with high parasitemia. Most of the 11 identified proteins are parasitic surface proteins that play a role in entry to the host pathway. This aligns with the results from a published pediatric parasitic transcriptomic analysis conducted in our group (Guillochon et al, 2022).

Altogether, these results revealed a specific protein profile of severe malaria in human plasma, along with an increase in the circulating 20S proteasome, consistent with a previously published study (Mahamar et al, 2021). Moreover, a highly dysregulated iron metabolism pathway and an increased inflammation were identified in SM. The observed elevation of *TFRC* levels, specifically in iEs from the CM group, suggests a potential preference of *P. falciparum* for invading reticulocytes or erythroid precursors. This preference could enable the parasite to boost its growth, resulting in a faster increase of parasite load within the human body. Such accelerated growth may play a pivotal role in the development of CM. Nevertheless, these findings require functional analysis of the molecular mechanisms underlying the invasion of reticulocytes or erythroid precursors by the parasite and contradict in vitro studies performed on laboratory strains (Ong et al, 2023). This deeper understanding could pave the way for the development of molecules aimed at preventing the invasion of these cells by *Plasmodium falciparum* to limit CM development. Our findings exclusively reflected the patient's health status at the onset of the severe symptoms and have the potential to serve as a foundation for comprehensive future research aimed at developing a predictive tool for diagnosing severe malaria in the field.

# Methods

## Study location

This study used the iE and plasma samples from three previous field studies conducted in the Cotonou region, Atlantic department, in southern Benin. Two independent patient enrolments were conducted to recruit malaria cases: CIVIC in 2016 (Kamaliddin et al, 2019) and NeuroCM in 2018 (Joste et al, 2019; Brisset et al, 2022). Patients were recruited from the Lagune Mother and Child Hospital in Cotonou, the Saint-Joseph Hospital in Sô-Ava, and the area hospital of Calavi. Malaria transmission in Benin is seasonal, with the highest transmission rates from April to August during the rainy season. A third study named Asympto2020 recruited noninfected (NI) children also in Benin.

## Inclusion criteria

These studies included patients who were under the age of five. Febrile patients were tested for *P. falciparum* infection using a rapid diagnostic test (HRP-2; SD Bioline, Yongin, South Korea), and those who tested positive were included in the infected patient group. A trained microscopist determined the density of *P. falciparum* by examining Giemsa-stained thick blood smears. Slides were considered negative after examining 8000 white blood cells. Species identification was confirmed using species-specific qPCR, and only patients with *P. falciparum* mono-specific infections were selected for downstream proteomic analysis (Kamaliddin et al, 2019). Clinical cases classification was based

 

on the following criteria: SM was defined as any severity criterion according to the WHO (WHO, 2013). CM was defined based on a coma (Blantyre score ≤2) and the absence of other infectious etiologies assessed by cerebrospinal fluid count and culture; SMA was defined by a hemoglobin level <5 g/dL, measured by Hemocue 201 device (Radiometer, Brønshøj, Denmark), and a Blantyre score >2. This study will use the term "SM" when analyzing the combination of CM and SMA samples. Children with UM presented with a fever without any severity criteria. In addition, non-infected (NI) Beninese children were included. These children were screened negative for malaria using Rapid Diagnostic Test (RDT) and confirmed by negative *Plasmodium spp/Pf/Po/Pm* PCR (Sandeu et al, 2012).

## Sample processing

Five milliliters of peripheral whole blood were collected from the patients using EDTA vacutaine tubes at the time of inclusion and prior to hospital treatment initiation. The plasma was separated from iE by centrifugation for 10 min at $2000 \times g$, and immediately stored at $-80\,°C$. The remaining cells were mixed with sterile PBS (Gibco - Thermo Fisher Scientific, Waltham, USA) in a 1:1 (vol:vol) ratio, and white blood cells were removed using a density gradient centrifugation (Lymphoprep; Axis-Shield PoC AS, Oslo, Norway). The obtained erythrocytes were then incubated in RPMI medium supplemented with human serum AB+ (PAA laboratories, Velizy-Villacoublay, France) and Albumax II (Gibco - Thermo Fisher Scientific, Waltham, USA) until parasite maturation to the late trophozoite stage (18–32 h). This controlled environment was designed to compensate for the predominantly young parasite stages found in circulating blood, which expressed only a limited number of surface proteins. The late trophozoites were subsequently isolated by magnetic activated cell sorting (MACS) (Myltenyi Biotech, Bergisch Gladbach, Germany), as previously described by (Ribaut et al, 2008) and preserved at $-80\,°C$.

## Mature iE samples LC-MS/MS analysis

The samples were prepared for LC-MS/MS following the methodology previously described (Kamaliddin et al, 2019). Initially, iE pellets were resuspended with 25 µL of solubilization buffer (1% sodium desoxycholate, 100 mM Tris/HCl pH 8.5, 10 mM tris(2-carbox-yethyl)phosphine (TCEP), 40 mM chloroacetamide). The iE resuspension was heated at 95 °C for 5 min and subjected to three rounds of sonication lasting 30 s each. The extracts were diluted (1:1; v/v) in Tris-acetonitrile (ACN) buffer (50 mM Tris/HCl pH 8.5, 10% ACN) and digested overnight at 37 °C using 1 µg of trypsin (V5111; Promega, Madison, USA) for 50 µg of proteins. The resulting peptides were fractionated into five fractions per sample by strong cationic exchange (SCX) as previously described (Kulak et al, 2014), and further dried using a SpeedVac. Each SCX fraction was solubilized in 10% ACN and 0.1% trifluoroacetic acid (TFA) and loaded into a Dionex U3000 RSLC nano-LC-system (2 µm particles, C18 reverse phase column, 15 cm length, 75 µm inner diameter from Dionex, Sunnyvale, USA) using a 180 min gradient program coupled to an Orbitrap-fusion mass spectrometer (Thermo Fisher Scientific, Waltham, USA). Samples were randomly injected into the mass spectrometer to avoid possible technical biases.

## Plasma samples LC-MS/MS analysis

Plasma samples were depleted of the 14 major plasma proteins using a targeted antibody-based depletion kit (Top 14 Abundant Protein Depletion Spin Columns, Thermo Fisher Scientific, Waltham, USA), following the manufacturer's instructions. Briefly, 10 µL of plasma was loaded onto each column containing an antibody-coated resin and incubated at room temperature on a rotator disk at low speed for one hour. After centrifugation at $1000 \times g$ for 2 minutes, the eluates of depleted plasma were collected and then centrifuged at $15\ 000\,g$ for 45 min through an Amicon 10 kDa filter concentrator. The resulting proteins were denatured, reduced, and alkylated using a denaturing buffer (6 M Urea, 100 mM Tris/HCl pH 8.5, 10 mM TCEP, and 50 mM chloroacetamide). The urea was diluted to 2 M in 100 mM Tris/HCl pH 8.5, thus permitting trypsin activity (V5111; Promega, Madison, USA) at 37 °C overnight (1 µg trypsin for 50 µg of protein).

The following day, peptides were dried in a SpeedVac, resuspended in 500 µL of Buffer A (10 mM ammonium formate, pH 10 in mQ $H_2O$) and sonicated for five minutes before being loaded onto an AKTA chromatography system equipped with a Zorbax Extend-$C_{18}$ column (Agilent, Santa Clara, USA) for Reverse-Phase High pH (RP-HpH) peptide fractionation. Peptides bound to the hydrophobic C18 phase were eluted with Buffer B (10 mM ammonium formate, pH 10, in 80% acetonitrile). Next, 250 µL fractions were collected at a flow rate of 0.250 mL/min during 1h24min (total vol. = 21 mL) on three different gradients (Buffer A and B mixed) with an initial (from 0 to 2 mL) flow rate at 6.25% of buffer B (93.75% of Buffer A): (1) 6.25–44% of Buffer B (from 2 mL to 14.5 mL), (2) 44–75% of Buffer B (from 14.5 mL to 17 mL), and (3) 75–100% of Buffer B (from 17 mL to 21 mL). The 84 harvested fractions were pooled into nine final fractions based on a concatenation plan to equalize the amount of peptides in each fraction (Appendix Fig. S5).

The nine fractions were subsequently dried using a SpeedVac. Peptides from each RP-HpH fraction were solubilized in 25 µL of 10% ACN and 0.1% TFA, and 5 µL corresponding to 1 µg of peptides were injected. A data-dependent analysis (DDA) with PASEF enabled method was set up by using Data Analysis and timsControl software (Bruker, Billerica, USA) and peptides from each fraction were separated into a $C_{18}$ reverse phase column (1.6 µm particles size Aurora, 75 µm inner diameter and 25 cm length from IonOptics) on a Dionex U3000 nLC system during a 2-h gradient program and electrosprayed into a timsTOF Pro mass spectrometer (Bruker, Billerica, USA). Plasma samples were randomly injected into the mass spectrometer to avoid possible technical biases.

## LC-MS/MS raw data analysis

The raw data obtained were analyzed using MaxQuant 1.6.6.0 for iE and MaxQuant 2.0.3 for plasma (Tyanova et al, 2016a), which queried the Uniprot/Swissprot sequence database for human proteins and PlasmoDB (v56) for *P. falciparum* 3D7 strain proteins (Alvarez-Jarreta et al, 2023). The protein identification false discovery rate (FDR) was set to 1%, with a match between runs enabled, and identification was based on unique + razor peptides with at least one peptide. A minimum of two ratios of unique +

razor peptides was required for quantification. Protein quantification analysis was performed using Label-Free Quantification (LFQ) intensities data calculated by the MaxQuant algorithm. The MaxQuant files were further analyzed using Perseus software (Tyanova et al, 2016b). Data cleanup was performed by eliminating the (i) reverse protein sequence, (ii) proteins identified only by one modification site and (iii) potential contaminants based on the contaminant list from MaxQuant. The resulting proteins LFQ values were $\log2(x)$ transformed and importantly a minimum of 65% of available quantitative values in at least one group was needed to include proteins in the statistical analysis. For example, in plasma 4 LFQ values (out of 6 total samples in a group) in at least one clinical group are required to include the protein.

## LC-MS/MS validation and iron metabolism biomarker quantification

For quantification, 340 plasma samples from the CIVIC, NeuroCM and Asympto2020 cohorts were used. We selected a subset of three proteins highly differentially abundant and well-characterized (*CRP*, *FTL*, and *HPX*) to confirm the mass spectrometry quantification values. Other markers of iron metabolism and inflammation pathways that could not be acquired by LC-MS/MS (depleted or not enough quantification values) were also quantified. Briefly, C-reactive protein (*CRP*), Ferritin (*FTL*), Albumin (*ALB*), Haptoglobin (*HP*), total and conjugated Bilirubin (t*BIL* and c*BIL*) and Transferrin (*TF*) were measured on a Cobas Pro (Roche, Basel, Switzerland) as recommended by the manufacturer. Hemopexin (*HPX*) was measured on an Atellica Neph (Siemens Healthineers, Erlangen, Germany).

Hepcidin concentrations in plasma were quantified by ELISA assay using a Quantikine ELISA kit (DHP250; R&D system, Minneapolis, USA), following the manufacturer's guidelines. Absorbance values (OD450nm) were measured on an EnSpire system (PerkinElmer, Waltham, USA) and OD570nm values were subtracted.

Transferrin receptor protein 1 (TFRC) concentrations in infected erythrocytes whole cell lysate were quantified by ELISA following the manufacturer's guidelines (A4497, antibodies.com, Sweden). Absorbance values (OD450nm) were measured on a Multiskan FC plate reader (Thermo Fisher Scientific, Waltham, USA), and samples concentrations were calculated using a linear regression curves.

## Over-representation analysis (ORA) and interaction network of the deregulated pathways

The functional significance of the differentially abundant host proteins observed in the different clinical groups, was analyzed with Ingenuity Pathways Analysis (IPA; Qiagen, Hilden, Germany). For plasma analysis, ANOVA $q$-value < 0.05 and Log2 Foldchange <−1 and >1 were used as cutoffs. For iE, no foldchange cut-off filters were used. Fisher's exact test was employed as the statistical test to identify enriched canonical pathways and the IPA algorithm (based on $Z$-score) was used to determine whether a pathway was activated or inhibited. Interaction networks were created using the classical and sub-cellular models of IPA to obtain an overview of the localization of deregulated proteins.

The differentially abundant parasitic proteins (for which IPA was not applicable) identified between the groups were analyzed using the PANTHER online tool (http://www.pantherdb.org/), which combines the three primary components of the Gene Ontology (GO): biological processes (BP), cellular component (CC), and molecular function (MF). Hypergeometric distribution tests were performed to determine the overlap of our gene sets. FDR $q$-value < 0.05 was reported as significantly over-represented for multiple hypothesis testing using the Benjamini–Hochberg method. The interaction networks of the differentially abundant proteins identified in the LC-MS/MS analysis and the over-representation analysis (ORA) for *P. falciparum* proteins were generated using the STRING online tool (https://string-db.org/).

## Statistical analysis

For plasma samples, ANOVA analysis was performed with Benjamini–Hochberg correction. The host's proteins with a $q$-value < 0.05 were defined as significantly deregulated. Post hoc Tukey's HSD test was then carried out on ANOVA significant proteins to determine which groups exhibit deregulated proteins.

For iE samples, Student's $t$ tests ($p < 0.05$) without Benjamini–Hochberg correction were performed due to a lack of CM samples ($n = 3$) and lower quantification coverage. For graphical representations, the differentially abundant proteins (DAP) missing LFQ values were imputed based on the normal distribution (Width = 0.3 and Down shift = 1.8), and all values were normalized with $Z$-score ($Z = \frac{X - \mu}{\sigma}$) (where $X$ is the sample log2LFQ value, $\mu$ the mean of all log2LFQ values for a protein, and $\sigma$ the standard deviation). Then, $Z$-score values were used to build heatmaps using Euclidean hierarchical clustering. To compare the abundance of plasmatic biomarkers obtained through ELISA-based assays, Mann–Whitney $U$-tests were performed using R software to calculate the exact $p$ values, and proteins with a $p$ value < 0.05 were considered significantly dysregulated. The operator for statistical analyses was not blinded to experimental conditions and sample ID.

## Study approval

The study obtained ethical clearance from the Institutional Ethics Committee of the Faculty of Health Science at the Abomey-Calavi University in Benin (clearance no.90, 06/06/2016) for the CIVIC cohort. Written informed consent was obtained from the guardians of the children before their inclusion in the study by trained study staff. The study did not interfere with patient care and treatment and patients were treated following with the national malaria program policy. For the NeuroCM cohort, ethical clearance was obtained from the Comité National d'Ethique pour la recherche en santé au Benin (No.67/MS/DC/SGM/DRFMT/CNERS/SA; 17 October 2017), and the field study protocol was previously described (Joste et al, 2019). For the NI children (Asympto2020) ethical clearance from the Comité d'Ethique Institutionnel du CREC (No.023/CREC/CEI-CREC/SA) was obtained. The study was carried out following the WMA Declaration of Helsinki and the Department of Health and Human Services Belmont Report, adhering to ethical principles for medical research involving human subjects.

**The paper explained**

**Problem**

Severe cases of *P. falciparum* malaria led to the deaths of over 488,391 children under 5 years old in 2022. Despite extensive research, particularly focused on parasitic factors, the precise mechanisms underlying these severe cases in patients remain unclear. Both the host and parasite proteomes may contribute to deciphering pediatric severe malaria.

**Results**

We screened the proteomes of parasitized red blood cells and plasma using mass spectrometry in a cohort of children under 5 years old from Benin presenting various clinical presentations of malaria. In the plasma, we identified a protein signature of severe malaria primarily associated with host proteins, highlighting the significant role of the host "response to infection" in severe malaria. In the proteome of parasitized red blood cells, we also observed an increased transferrin receptor abundance, suggesting a more significant invasion of reticulocytes or erythroblasts in patients experiencing cerebral malaria.

**Impact**

Our study provides a better understanding of *P. falciparum* malaria and the associated host response at the protein level. It paves the way for developing new diagnostic tools to predict disease worsening and thus better target children at high risk of complication.

## Data availability

Proteomic acquired raw data for plasma and iE samples are available on ProteomeXchange via the PRIDE repository (https://www.ebi.ac.uk/pride): mass spectrometry raw data for the plasma proteome with the identifier PXD043033; mass spectrometry raw data for the infected erythrocyte proteome with the identifier PXD042503.

## Peer review information

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

© This is a U.S. Government work and not under copyright protection in the US; foreign copyright protection may apply

*Plasmodium falciparum* and host factors associated with cerebral malaria: description of the protocol for a prospective, case-control study in Benin (NeuroCM). BMJ Open 9:e027378

Kaestli M, Cockburn IA, Cortés A, Baea K, Rowe JA, Beck H-P (2006) Virulence of malaria is associated with differential expression of *Plasmodium falciparum* var gene subgroups in a case-control study. J Infect Dis 193:1567–1574

Kamaliddin C, Rombaut D, Guillochon E, Royo J, Ezinmegnon S, Agbota G, Huguet S, Guemouri S, Peirera C, Coppée R et al (2019) From genomic to LC-MS/MS evidence: analysis of PfEMP1 in Benin malaria cases. PLoS ONE 14:e0218012

Klausner RD, Rouault TA, Harford JB (1993) Regulating the fate of mRNA: The control of cellular iron metabolism. Cell 72:19–28

Krause RGE, Hurdayal R, Choveaux D, Przyborski JM, Coetzer THT, Goldring JPD (2017) Plasmodium glyceraldehyde-3-phosphate dehydrogenase: a potential malaria diagnostic target. Exp Parasitol 179:7–19

Kulak NA, Pichler G, Paron I, Nagaraj N, Mann M (2014) Minimal, encapsulated proteomic-sample processing applied to copy-number estimation in eukaryotic cells. Nat Methods 11:319–324

Kumar V, Ray S, Aggarwal S, Biswas D, Jadhav M, Yadav R, Sabnis SV, Banerjee S, Talukdar A, Kochar SK et al (2020) Multiplexed quantitative proteomics provides mechanistic cues for malaria severity and complexity. Commun Biol 3:683

Lavstsen T, Turner L, Saguti F, Magistrado P, Rask TS, Jespersen JS, Wang CW, Berger SS, Baraka V, Marquard AM et al (2012) *Plasmodium falciparum* erythrocyte membrane protein 1 domain cassettes 8 and 13 are associated with severe malaria in children. Proc Natl Acad Sci USA 109:E1791–1800

Mahamar A, Gonzales Hurtado PA, Morrison RD, Boone R, Attaher O, Diarra BS, Gaoussou S, Issiaka D, Dicko A, Duffy PE et al (2021) Plasma biomarkers of hemoglobin loss in *Plasmodium falciparum*-infected children identified by quantitative proteomics. Blood 139:2361–2376

Matzner Y, Hershko C, Polliack A, Konijn AM, Izak G (1979) Suppressive effect of ferritin on in vitro lymphocyte function. Br J Haematol 42:345–353

Menendez C, Fleming AF, Alonso PL (2000) Malaria-related anaemia. Parasitol Today Pers Ed 16:469–476

Milner DA (2018) Malaria pathogenesis. Cold Spring Harb Perspect Med 8:a025569

Moura IC, Hermine O, Lacombe C, Mayeux P (2015) Erythropoiesis and transferrin receptors. Curr Opin Hematol 22:193–198

Nemeth E, Tuttle MS, Powelson J, Vaughn MB, Donovan A, Ward DM, Ganz T, Kaplan J (2004) Hepcidin regulates cellular iron efflux by binding to ferroportin and inducing its internalization. Science 306:2090–2093

Neveu G, Richard C, Dupuy F, Behera P, Volpe F, Subramani PA, Marcel-Zerrougui B, Vallin P, Andrieu M, Minz AM et al (2020) *Plasmodium falciparum* sexual parasites develop in human erythroblasts and affect erythropoiesis. Blood 136:1381–1393

Novelli EM, Hittner JB, Davenport GC, Ouma C, Were T, Obaro S, Kaplan S, Ong'echa JM, Perkins DJ (2010) Clinical predictors of severe malarial anaemia in a holoendemic *Plasmodium falciparum* transmission area. Br J Haematol 149:711–721

O'Donnell A, Fowkes FJI, Allen SJ, Imrie H, Alpers MP, Weatherall DJ, Day KP (2009) The acute phase response in children with mild and severe malaria in Papua New Guinea. Trans R Soc Trop Med Hyg 103:679–686

Ong JJY, Oh J, Yong Ang X, Naidu R, Chu TTT, Hyoung Im J, Manzoor U, Kha Nguyen T, Na S-W, Han E-T et al (2023) Optical diffraction tomography and image reconstruction to measure host cell alterations caused by divergent Plasmodium species. Spectrochim Acta A Mol Biomol Spectrosc 286:122026

Pasvol G, Weatherall DJ, Wilson RJ (1980) The increased susceptibility of young red cells to invasion by the malarial parasite *Plasmodium falciparum*. Br J Haematol 45:285–295

Perkins DJ, Were T, Davenport GC, Kempaiah P, Hittner JB, Ong'echa JM (2011) Severe malarial anemia: innate immunity and pathogenesis. Int J Biol Sci 7:1427–1442

Rauch JN, Gestwicki JE (2014) Binding of human nucleotide exchange factors to heat shock protein 70 (Hsp70) generates functionally distinct complexes in vitro. J Biol Chem 289:1402–1414

Ribaut C, Berry A, Chevalley S, Reybier K, Morlais I, Parzy D, Nepveu F, Benoit-Vical F, Valentin A (2008) Concentration and purification by magnetic separation of the erythrocytic stages of all human Plasmodium species. Malar J 7:45

Sae-Lee W, McCafferty CL, Verbeke EJ, Havugimana PC, Papoulas O, McWhite CD, Houser JR, Vanuytsel K, Murphy GJ, Drew K et al (2022) The protein organization of a red blood cell. Cell Rep 40:111103

Sandeu MM, Moussiliou A, Moiroux N, Padonou GG, Massougbodji A, Corbel V, Ndam NT (2012) Optimized Pan-species and speciation duplex real-time PCR assays for Plasmodium parasites detection in malaria vectors. PLoS ONE 7:e52719

Seydel KB, Kampondeni SD, Valim C, Potchen MJ, Milner DA, Muwalo FW, Birbeck GL, Bradley WG, Fox LL, Glover SJ et al (2015) Brain swelling and death in children with cerebral malaria. N Engl J Med 372:1126–1137

Sharma P, Tóth V, Hyland EM, Law CJ (2021) Characterization of the substrate binding site of an iron detoxifying membrane transporter from *Plasmodium falciparum*. Malar J 20:295

Shonhai A, Blatch GL (2021) Heat shock proteins of malaria: highlights and future prospects. Adv Exp Med Biol 1340:237–246

Smalley ME, Abdalla S, Brown J (1981) The distribution of *Plasmodium falciparum* in the peripheral blood and bone marrow of Gambian children. Trans R Soc Trop Med Hyg 75:103–105

Spottiswoode N, Duffy PE, Drakesmith H (2014) Iron, anemia and hepcidin in malaria. Front Pharmacol 5:125

Srivastava A, Creek DJ, Evans KJ, Souza DD, Schofield L, Müller S, Barrett MP, McConville MJ, Waters AP (2015) Host reticulocytes provide metabolic reservoirs that can be exploited by malaria parasites. PLoS Pathog 11:e1004882

Srivastava A, Evans KJ, Sexton AE, Schofield L, Creek DJ (2017) Metabolomics-based elucidation of active metabolic pathways in erythrocytes and HSC-derived reticulocytes. J Proteome Res 16:1492–1505

Storm J, Craig AG (2014) Pathogenesis of cerebral malaria—inflammation and cytoadherence. Front Cell Infect Microbiol 4:100

Thomson-Luque R, Votborg-Novél L, Ndovie W, Andrade CM, Niangaly M, Attipa C, Lima NF, Coulibaly D, Doumtabe D, Guindo B et al (2021) *Plasmodium falciparum* transcription in different clinical presentations of malaria associates with circulation time of infected erythrocytes. Nat Commun 12:4711

Tonkin-Hill GQ, Trianty L, Noviyanti R, Nguyen HHT, Sebayang BF, Lampah DA, Marfurt J, Cobbold SA, Rambhatla JS, McConville MJ et al (2018) The *Plasmodium falciparum* transcriptome in severe malaria reveals altered expression of genes involved in important processes including surface antigen–encoding var genes. PLoS Biol 16:e2004328

Tyanova S, Temu T, Cox J (2016a) The MaxQuant computational platform for mass spectrometry-based shotgun proteomics. Nat Protoc 11:2301–2319

Tyanova S, Temu T, Sinitcyn P, Carlson A, Hein MY, Geiger T, Mann M, Cox J (2016b) The Perseus computational platform for comprehensive analysis of (prote)omics data. Nat Methods 13:731–740

Wahlgren M, Goel S, Akhouri RR (2017) Variant surface antigens of *Plasmodium falciparum* and their roles in severe malaria. Nat Rev Microbiol 15:479–491

Wang W, Knovich MA, Coffman LG, Torti FM, Torti SV (2010) Serum ferritin: past, present and future. Biochim Biophys Acta 1800:760–769

Weiss G, Ganz T, Goodnough LT (2019) Anemia of inflammation. Blood 133:40–50

WHO (2013) Management of severe malaria: a practical handbook. WHO, Geneva

WHO (2022) World malaria report. WHO, Geneva

Yan Q, Sun W, Kujala P, Lotfi Y, Vida TA, Bean AJ (2005) CART: an Hrs/actinin-4/BERP/myosin V protein complex required for efficient receptor recycling. Mol Biol Cell 16:2470–2482

## Acknowledgements

The authors thank all patients and their guardians for their kind participation in the study and all nurses and laboratory staff from the Hôpital de zone de Calavi, CHU-MEL and St-Joseph hospitals involved in patient's recruitment. The authors would like to express a special thanks to Patrick Mayeux for his critical help in the development of the high pH chromatography method on AKTA device. We also want to thank Patrick Fraering for his critical support in reviewing the manuscript, Nadine Fievet for managing the field study workers, Gilles Cottrell and the CERPAGE lab for NI plasma collection sharing, Morgane Le Gall for her crucial help with IPA software, and the NeuroCM consortium (detailed at the end of the paper) for the validation cohort plasma sample sharing. The authors would also like to thank Cédric Broussard, Marjorie Leduc, and Johanna Bruce from the Proteom'IC core facility (Institut Cochin) for their critical technical help during the whole project duration. We also thank the funders of the mass spectrometers: IbiSa, DIM Thérapie Génique, Gr-Ex Labex, Université Paris Cité, Canceropôle Ile-de-France, l'Association de Prévention et d'Etudes des Maladies Moléculaires (APEMM) and le fond européen de développement régional (FEDER). This study was also funded by a Mérieux Research Grant to GIB (http://www.institut-merieux.com/fr/accueil/) and a PhD Scholarship from the French Minister of Research to CK (ED 563, Université Paris Cité) awarded to CK for the work on the initial CIVIC cohort. A grant from ANR-17-CE17-0001 to GIB funded the NeuroCM project. The funders had no role in study design, data collection and analysis, decision to publish, or manuscript preparation.

## Author contributions

**Jérémy Fraering**: Conceptualization; Software; Formal analysis; Validation; Investigation; Visualization; Methodology; Writing—original draft; Writing—review and editing. **Virginie Salnot**: Data curation; Software; Methodology; Writing—review and editing. **Emilie-Fleur Gautier**: Resources; Data curation; Software; Supervision; Methodology; Writing—review and editing. **Sem Ezinmegnon**: Resources; Investigation. **Nicolas Argy**: Investigation; Writing—review and editing. **Katell Peoc'h**: Resources; Investigation; Methodology; Writing—review and editing. **Hana Manceau**: Resources; Investigation; Methodology. **Jules Alao**: Resources. **François Guillonneau**: Software; Supervision; Methodology; Writing—review and editing. **Florence Migot-Nabias**: Supervision; Writing—review and editing. **Gwladys I Bertin**: Conceptualization; Resources; Data curation; Supervision; Funding acquisition; Project administration; Writing—review and editing. **Claire Kamaliddin**: Conceptualization; Data curation; Supervision; Funding acquisition; Visualization; Project administration; Writing—review and editing.

## Disclosure and competing interests statement

The authors declare no competing interests.

## NEUROCM CONSORTIUM

Agnes Aubouy[14], Dissou Affolabi[15], Bibiane Biokou[10], Michel Cot[1], Jean-Eudes Degbelo[16], Philippe Deloron[1], Latifou Dramane[16], Jean-François Faucher[17], Emilie Guillochon[1], Ludivine Houze[1], Sayeh Jafari-Guemouri[1], Sandrine Houze[6], Valentin Joste[1], Anaïs Labrunie[18], Yélé Ladipo[10], Thomas Lathiere[19], Achille Massougbodji[16], Audrey Mowendabeka[20], Jade Papin[1], Bernard Pipy[15], Pierre-Marie Preux[18], Marie Raymondeau[18], Jade Royo[15], Darius Sossou[16], Brigitte Techer[1] & Bertin Vianou[16]

[14]PHARMADEV, Université de Toulouse, IRD, UPS, Paris, France. [15]Pediatric Department, Calavi Hospital, Calavi, Benin. [16]Institut de Recherche Clinique du Bénin [IRCB], Calavi, Benin. [17]INSERM, Univ. Limoges, CHU Limoges, IRD, U1094 Tropical Neuroepidemiology, Institute of Epidemiology and Tropical Neurology, GEIST, Limoges, France. [18]EPIMACT, INSERM, Université de Limoges, Limoges, France. [19]Ophthalmology Department, Limoges University Hospital, Limoges, France. [20]Paediatric Department, Hopital de la Mère et de l'Enfant, Limoges, France.

# Expanded View Figures

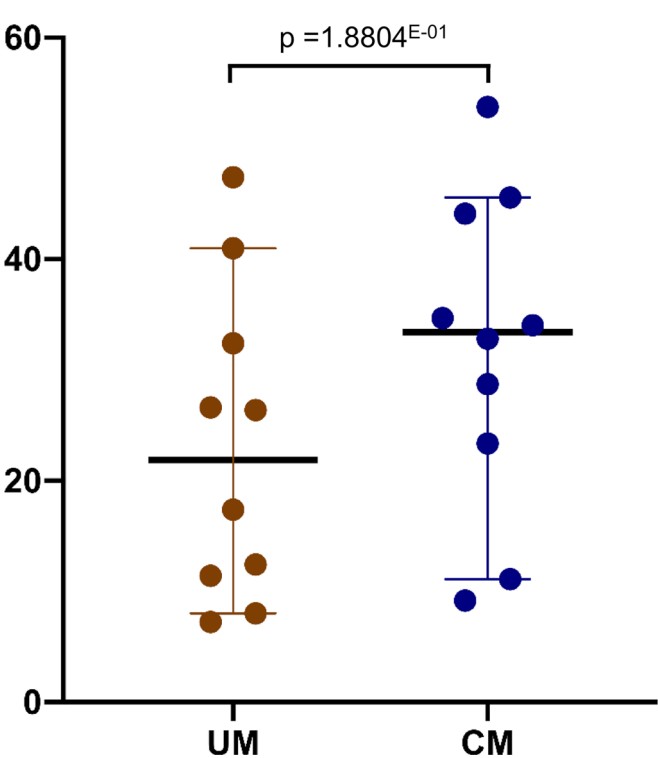

**Figure EV1. Targeted quantification of transferrin receptor protein 1 (*TFRC*) by ELISA assay.**

Dot plot representing *TFRC* concentration of 20 iE samples from the NeuroCM cohort (10 UM vs 10 CM) measured by ELISA assay. Data were analyzed with GraphPad Prism 8.0. Bold horizontal bars correspond to the median and colored bars correspond to the 95% CI. Mann–Whitney *U*-test was used as statistical test and *p* value is displayed in the figure. Source data are available online for this figure.

