## [Peer Review File · EMBO Molecular Medicine]

Infected erythrocytes and plasma proteomics reveals a specific protein signature of severe malaria

Jérémy Fraering, Virginie Salnot, Emilie-Fleur Gautier, Sem Ezinmegnon, Nicolas Argy, Katell Peoc'h, Hana Manceau, Jules Alao, François Guillonnet, Florence Migot-Nabias, Gwladys Bertin, and Claire Kamaliddin

DOI: [10.15252/emmm.202318226](https://doi.org/10.15252/emmm.202318226)

Corresponding authors: *Gwladys Bertin* (gwladys.bertin@ird.fr) , *Claire Kamaliddin* (claire.kamaliddin@ucalgary.ca)

Review Timeline:

Submission Date:	27th Jun 23
Editorial Decision:	27th Jul 23
Revision Received:	23rd Oct 23
Editorial Decision:	9th Nov 23
Revision Received:	18th Nov 23
Accepted:	22nd Nov 23

Editor: *Lise Roth*

Transaction Report:

27th Jul 2023

Dear Dr. Bertin,

Thank you for submitting your work to EMBO Molecular Medicine. We have now heard back from the referees who agreed to evaluate your manuscript. As you will see below, the reviewers raise substantial concerns on your work, which unfortunately preclude its publication in EMBO Molecular Medicine in its current form.

The reviewers find that the question addressed by the study is of potential interest, however they remain unconvinced that some of the major conclusions are sufficiently supported by the data. They thus raise the following major issues:

- lack of clarity throughout (methods, manuscript organization, grammar/syntax...)
- statistical analyses (corrections for several tests required)

If you feel you can satisfactorily address these points and those listed by the referees, you may wish to submit a revised version of your manuscript. Please attach a covering letter giving details of the way in which you have handled each of the points raised by the referees. A revised manuscript will once again be subject to review, and we cannot guarantee at this stage that the eventual outcome will be favorable.

We are expecting your revised manuscript within three months. If you anticipate any delay, please contact us.

We require:

4) A .docx formatted letter INCLUDING the reviewers' reports and your detailed point-by-point responses to their comments. As part of the EMBO Press transparent editorial process, the point-by-point response is part of the Review Process File (RPF), which will be published alongside your paper.

5) A complete author checklist, which you can download from our author guidelines (<https://www.embopress.org/page/journal/17574684/authorguide#submissionofrevisions>). Please insert information in the checklist that is also reflected in the manuscript. The completed author checklist will also be part of the RPF.

6) It is mandatory to include a 'Data Availability' section after the Materials and Methods. Before submitting your revision, primary datasets produced in this study need to be deposited in an appropriate public database, and the accession numbers and database listed under 'Data Availability'. Please remember to provide a reviewer password if the datasets are not yet public (see <https://www.embopress.org/page/journal/17574684/authorguide#dataavailability>).

7) For data quantification: please specify the name of the statistical test used to generate error bars and P values, the number (n) of independent experiments (specify technical or biological replicates) underlying each data point and the test used to calculate p-values in each figure legend. The figure legends should contain a basic description of n, P and the test applied. Graphs must include a description of the bars and the error bars (s.d., s.e.m.). Please provide exact p values.

8) Our journal encourages inclusion of *data citations in the reference list* to directly cite datasets that were re-used and obtained from public databases. Data citations in the article text are distinct from normal bibliographical citations and should directly link to the database records from which the data can be accessed. In the main text, data citations are formatted as follows: "Data ref: Smith et al, 2001" or "Data ref: NCBI Sequence Read Archive PRJNA342805, 2017". In the Reference list, data citations must be labeled with "[DATASET]". A data reference must provide the database name, accession

number/identifiers and a resolvable link to the landing page from which the data can be accessed at the end of the reference. Further instructions are available at .

9) We replaced Supplementary Information with Expanded View (EV) Figures and Tables that are collapsible/expandable online. A maximum of 5 EV Figures can be typeset. EV Figures should be cited as 'Figure EV1, Figure EV2' etc... in the text and their respective legends should be included in the main text after the legends of regular figures.

10) The paper explained: EMBO Molecular Medicine articles are accompanied by a summary of the articles to emphasize the major findings in the paper and their medical implications for the non-specialist reader. Please provide a draft summary of your article highlighting

11) For more information: There is space at the end of each article to list relevant web links for further consultation by our readers. Could you identify some relevant ones and provide such information as well? Some examples are patient associations, relevant databases, OMIM/proteins/genes links, author's websites, etc...

12) Author contributions: CRediT has replaced the traditional author contributions section because it offers a systematic machine readable author contributions format that allows for more effective research assessment. Please remove the Authors Contributions from the manuscript and use the free text boxes beneath each contributing author's name in our system to add specific details on the author's contribution. More information is available in our guide to authors.

13) Disclosure statement and competing interests: We updated our journal's competing interests policy in January 2022 and request authors to consider both actual and perceived competing interests. Please review the policy <https://www.embopress.org/competing-interests> and update your competing interests if necessary.

14) Every published paper now includes a 'Synopsis' to further enhance discoverability. Synopses are displayed on the journal webpage and are freely accessible to all readers. They include a short stand first (maximum of 300 characters, including space) as well as 2-5 one-sentences bullet points that summarizes the paper. Please write the bullet points to summarize the key NEW findings. They should be designed to be complementary to the abstract - i.e. not repeat the same text. We encourage inclusion of key acronyms and quantitative information (maximum of 30 words / bullet point). Please use the passive voice. Please attach these in a separate file or send them by email, we will incorporate them accordingly.

15) As part of the EMBO Publications transparent editorial process initiative (see our Editorial at <http://embomolmed.embopress.org/content/2/9/329>), EMBO Molecular Medicine will publish online a Review Process File (RPF) to accompany accepted manuscripts.

In the event of acceptance, this file will be published in conjunction with your paper and will include the anonymous referee reports, your point-by-point response and all pertinent correspondence relating to the manuscript. Let us know whether you agree with the publication of the RPF and as here, if you want to remove or not any figures from it prior to publication. Please note that the Authors checklist will be published at the end of the RPF.

I look forward to receiving your revised manuscript.

Yours sincerely,

Lise Roth

***** Reviewer's comments *****

Referee #1 (Comments on Novelty/Model System for Author):

The models chosen and questions asked are appropriate.

Referee #1 (Remarks for Author):

In their manuscript, Fraering et al describes the plasma proteome signatures associated with Severe Malaria and Cerebral Malaria, caused by *P. falciparum* infection. Through quantitative non-targeted proteomics, they report that in patients with Cerebral Malaria, infected erythrocytes contain higher abundance of transferrin receptor, indicative of preferential invasion of reticulocytes by *P. falciparum*. Overall, this study is robustly designed, manuscript well -presented and conclusions backed by data, I have only minor comments.

1. Did the authors check the infected erythrocytes for expression of CD71 or any other markers (CD98 protein or erythrocyte RNA through sub-vital NMB staining) indicative of young red cells ? This may provide direct validation of the proteomic results suggesting that parasite-infected RBCs are predominantly immature RBCs/reticulocytes. Obviously, such an experiment need to be done with ring stage infections directly (prior to allowing them to mature in culture) because the markers of immature reticulocytes tend to diminish within 20 hours of parasite infection.
2. Adding to the previous comment, did the authors observe higher number of uninfected reticulocytes in SM and/or CM samples? This was briefly mentioned in the discussion, but may need to be expanded. If there are infected and uninfected reticulocytes in a culture, the infected ones seem to mature faster than the uninfected ones according to recent literature (Ong et al, Spectrochimica Acta 2023). Hence, the argument that non-infected reticulocytes mature and infected ones remain the same as the authors suggest, are not in agreement.
3. Regarding accelerated growth of the parasites in CM condition as proposed by the authors, it is conceivable that the parasites mature faster due to suitable nutrient environment provided by host reticulocytes they are within. However, are they able to provide experimental evidence for this- such as stage-specific distribution of parasites at the beginning (mostly ring) and end of in vitro maturation step (18-32 hours later, when they should be trophozoites/schizonts), between the samples tested ?
4. The manuscripts have several typos throughout, and needs to be corrected in a revision.

Referee #2 (Comments on Novelty/Model System for Author):

My major concern refers to the analysis and conclusions derived from infected erythrocytes (iE). Student's t-tests ($p < 0.05$) without Benjamini-Hochberg correction were performed due to a lack of CM samples ($n = 3$) and lower quantification coverage. This raises doubts about the robustness of the differences between groups (high risk of false discovery rate). The conclusion about the preferential infection of reticulocytes or erythroid precursors by parasites causing CM, and the alteration of their maturation, is commented extensively in the discussion, without considering the potential limitation due to limited sample size. The wording used is very conclusive (ie, 648: "This preference enables the parasite to boost its growth, resulting in a faster increase of parasite load within the human body") and not sustained by the data. I recommend tuning down this part. A similar issue of potential spurious results should be considered for the p values presented in Figure 4, which should be corrected for multiple comparisons between groups. Some of the observation (ie, $p = 0.035$) have a high risk of being chance findings.

Referee #2 (Remarks for Author):

The manuscript by Fraering et al intitled "Infected erythrocytes and plasma proteomics reveals a specific protein signature of severe malaria" uses an unbiased proteomic assessment of infected erythrocytes ($n=17$) and plasma samples ($n=24$) collected from Beninese children, followed by targeted assays, to identify proteins differentially expressed between children with cerebral malaria (CM), severe anemia (SAM) and uncomplicated malaria (UM) as well as uninfected children. Up-regulation of the erythroid precursor marker transferrin receptor protein 1 (TFRC) was observed in CM infected erythrocytes. Increased levels of the 20S proteasome components were observed in plasmas from children with SM. Findings were confirmed using targeted assays in a cohort of 346 children. Authors conclude that the results suggest a) that parasites causing CM preferentially infect

reticulocytes or erythroid precursors and alter their maturation, and b) that the host plasma proteome serves as a specific signature of SM which may be used to develop novel diagnostic and prognostic biomarkers.

The manuscript, which requires some grammar/syntaxes check, is very dense, repetitive and difficult to follow in several sections. The conclusions are very speculative. A higher synthesis, especially in the introduction and discussion, may increase the clarity of the paper. As stated in the abstract, the objective is highly ambitious ("to decipher the complex mechanisms underlying the pathophysiology of CM and the corresponding host-parasite interactions") and difficult to achieve with the limited number of samples included in the study.

My major concern refers to the analysis and conclusions derived from infected erythrocytes (iE). Student's t-tests ($p < 0.05$) without Benjamini-Hochberg correction were performed due to a lack of CM samples ($n = 3$) and lower quantification coverage. This raises doubts about the robustness of the differences between groups (high risk of false discovery rate). The conclusion about the preferential infection of reticulocytes or erythroid precursors by parasites causing CM, and the alteration of their maturation, is commented extensively in the discussion, without considering the potential limitation due to limited sample size. The wording used is very conclusive (ie, 648: "This preference enables the parasite to boost its growth, resulting in a faster increase of parasite load within the human body") and not sustained by the data. I recommend tuning down this part.

A similar issue of potential spurious results should be considered for the p values presented in Figure 4, which should be corrected for multiple comparisons between groups. Some of the observation (ie, $p = 0.035$) have a high risk of being chance findings.

Sometimes it is difficult to see if the results presented refer to the infected erythrocytes or plasma samples.

Authors claim to have found a specific signature of severe malaria but they only included children with cerebral malaria and severe anemia. The use of severe malaria along the manuscript should be revised.

Line 308: "A total of 346 Beninese children were enrolled in this study". But this is not correct. Children were enrolled in a previous study, and samples were selected for this study. How was the selection done?

Grammar syntaxis: check "Samples preparation" (176), "this results" (631), was (54, instead of were) and other errors in the document. Please also revise dots vs comma to express decimals.

Line 179: the iE resuspension (not only the proteins) were heated.

Clarity of some sections can be increased by removing repetitions. For example, the use of Student t-test and the $p < 0.05$ is mentioned two times in 257 and 278; the use of "65% of quantitative values in at least one group" is mentioned 4 times (234, 335, 341, 345): the identification of L-lactate dehydrogenase (pLDH -PF3D7_1324900) in all SM samples mentioned 2 times (627 and 474). All this increases the complexity of the document and could be minimized by a clear description in the methods.

397: Authors mention that they "confirmed the increase of FTL and the decrease of HPX in CM initially found by LC-MS/MS". However, this was not observed in the colorimetric results, as authors say. So, authors should use another word instead of confirmation to avoid driving the reader to confusion (may be observed).

337: Authors mention that "the number of proteins in UM samples ($n = 725$) was significantly lower compared with SMA ($n = 1176$) and CM ($n = 824$) ($p = 0.007$; Kruskal-Wallis rank sum test)". May authors be referring to the median number of proteins per sample instead to the actual number of samples? Otherwise I do not understand how the statistical test was used.

340: "For the purpose of conducting differential analysis, 619 proteins shared among all clinical presentations were selected by using a filter of 65% quantitative values in at least one group". I understand from this sentence that authors are excluding from the analysis those proteins which are not expressed on one or more groups. Can this lead to exclude highly differentially expressed proteins (ie, those that are not expressed in particular groups but may be expressed in others)?

474: Authors show that pLDH was detected in all infected samples. Later (632), author mention that "This results aligns with the diagnostics performances of pLDH in RDTs, which are less sensitive than RDTs detecting other proteins such as HRP2 (80)" Why HRP2 was not detected and differentially expressed, taking into account the very different parasite densities between groups?

The discussion is very long and difficult to identify clear key messages. Avoiding the repetition of results (551-553, 627 and others), a more synthetic presentation of ideas and a less speculative discussion may increase clarity.

628: Something seems to be missing in this sentence: "Detection (79), which we identified in all SM plasma samples".

Referee #3 (Comments on Novelty/Model System for Author):

The authors used a well-established shotgun proteomics pipeline from sample preparation, data acquisition and analysis with an appropriate number of biological replicate analyses.

The medical impact stems from the potential identification of novel plasma biomarkers to better classify cases of severe malaria for earlier intervention, but how these protein signatures would be assayed in the field is not obvious.

Referee #3 (Remarks for Author):

To identify potential new biomarkers of severe malaria (SM), Fraering and colleagues report the proteomics analysis of proteins from infected erythrocytes (iE) and plasma obtained from clinical isolates of children under 5 diagnosed with cerebral malaria (CM) and severe malarial anemia (SMA) and compared with uncomplicated malaria (UM) and non-infected individuals (NI). The authors follow a well-established shotgun proteomics pipeline by digesting proteins into tryptic peptides, separating these offline (either using strong cation exchange for the iE samples or high pH reverse phase fractionation for the plasma samples), and analyzing the fractions by RPLC coupled to tandem mass spectrometry in data dependent acquisition mode. The MSMS datasets were searched against a combined *Plasmodium falciparum* and human protein database using MaxQuant and Perseus for protein identification and label-free quantitation. Validation studies using the plasma of 2 large patient cohorts were performed via colorimetric assays to target and quantify proteins significantly shown to be dysregulated in SM in the LCMS datasets.

The authors chose to only report the proteins in plasma and iE that were assessed to be differentially abundant by pair-wise comparisons with controls (Supplemental Table S1). It would have been beneficial to also include a table with the entire list of identified and quantified proteins in each of the replicate analyses for the 3 clinical conditions and control. Only reporting p-values and log2 does not allow for evaluating reproducibility and variation between replicate analyses. In addition, reporting the complete list of proteins might help better understand the "65% quantitative values in at least one group" filtering criteria applied to "select proteins for downstream analysis". This filtering step is not clearly explained in the Materials & Methods section, e.g., this sentence on page 10, lines 234-235, could use reworking "proteins with less cross-replicates values were filtered out (65% of quantitative values in at least one group)".

Furthermore, the rationale for selecting proteins for differential analysis seems flawed since the authors only selected proteins "shared among all clinical presentations" (page 14, line 340). From the figure legend, the PCA analyses in Fig 2D-E were performed on 284 proteins quantified in all 4 conditions x 6 biological replicates of plasma samples. Since the CM condition had 114 unique proteins (Venn Diagram in Fig2A), I'd argue that including the proteins reproducibly quantified in the CM samples might have helped separate CM from SMA better.

Throughout the manuscript, several consistency issues with references to figures or supplemental tables not matching the sentences in the Results sections will need to be fixed/clarified:

Pg 16, ln. 389-390: "we observed in CM plasma samples a significant ($q=0.017$) decrease of Hemopexin (HPX) abundance when compared to UM (FC=-4.44) and SMA (FC=-2.96) (Figure 4A)." While Hemopexin is indeed shown in Fig4A in the cluster of 21 proteins highly expressed in CM samples, hemopexin, along with all other proteins in this cluster, is not listed in the LC-MSMS supplemental table reporting differentially expressed proteins in plasma (Table S1B). The data for these proteins should be added to this supplemental table.

Pg 16, ln. 391-393: "we found a significant increase of ferritin light chain (FTL) abundance in CM samples, when compared to UM ($p=0.0001$; FC=7.50) and SMA ($p=0.002$; FC=5.58) (Supplemental Table S1B)". Quantitative data for "Ferritin light chain" is not reported in S1B but rather is part of a set of 6 "inflammation biomarkers" listed in Table S1D. Why are these proteins not listed as differentially expressed in S1B is confusing.

Pg. 16, ln. 410-412: "analyzing the plasmatic proteome of malaria patients (six NI, six UM, six SMA, and six CM) and found 68 DAP (q value<0.05) among the 781 selected proteins (Supplemental Table S1B)." Again, Table S1B, as provided, only contains 33 entries not 68.

The authors used colorimetric assays to establish the concentration of 9 proteins in plasma samples from their large cohorts (Table S2A, with boxplots for 5 proteins shown in Fig3C). While LC-MSMS data is available for hemopexin (Fig4A) and ferritin light chain (TableS1D), it is unclear whether the other proteins tested by colorimetry were detected at all in the proteomics analyses since the authors only provide tables for filtered/differentially expressed proteins. A side note: in Table S2A, colorimetric assays were performed on 355 plasma samples not 348 as written on line 395.

Overall, while sample preparation, data acquisition and processing are technically sound, how the data is organized and discussed does not follow a logical flow. The panels assembled for Fig3 and 4 are particularly confusing: for example, data for hemopexin's concentration in plasma based on colorimetric assay is shown in Fig3C yet the proteomics data showing its high abundance in CM samples (leading to the subsequent colorimetric validation) is shown as a heat map in Fig4A. As a matter of course, the alpha-numerical labeling of figure panels should follow the order each panel is being discussed in the main text.

In summary, I would strongly suggest the authors provide a supplemental table for the unfiltered quantitative protein lists for both the iE and plasma samples; tidy up their supplemental tables to match what is being stated in the main text; and reorganize results into sections that focus on 1 specific biological pathway with 1 figure containing panels with all relevant information for this group of proteins. This should help the readers make the most of this interesting study.

Below please find our point-point responses to the issues raised by reviewer:

Referee #1 (Comments on Novelty/Model System for Author): The models chosen and questions asked are appropriate.

Referee #1 (Remarks for Author):

In their manuscript, Fraering et al describes the plasma proteome signatures associated with Severe Malaria and Cerebral Malaria, caused by *P. falciparum* infection. Through quantitative non-targeted proteomics, they report that in patients with Cerebral Malaria, infected erythrocytes contain higher abundance of transferrin receptor, indicative of preferential invasion of reticulocytes by *P. falciparum*. Overall, this study is robustly designed, manuscript well -presented and conclusions backed by data, I have only minor comments.

1. Did the authors check the infected erythrocytes for expression of CD71 or any other markers (CD98 protein or erythrocyte RNA through sub-vital NMB staining) indicative of young red cells ? This may provide direct validation of the proteomic results suggesting that parasite-infected RBCs are predominantly immature RBCs/reticulocytes. Obviously, such an experiment need to be done with ring stage infections directly (prior to allowing them to mature in culture) because the markers of immature reticulocytes tend to diminish within 20 hours of parasite infection.

We thank the reviewer for this insightful suggestion. This work is the secondary analysis of a 2016 field study ("CIVIC project"), which was designed to compare protein abundance according to malaria severity. We agree with the reviewer on the importance of such experiments to confirm the hypothesis of early-stage erythrocytes preferential infection and confirm our findings from this manuscript. We no longer have any suitable biological material (slides) from the two cohorts (CIVIC and NeuroCM), and therefore cannot conduct any confirmatory study. So we only quantified the transferrin receptor protein 1 (TFRC or CD71) by LC-MS/MS in our study, we do not detect CD98. However, based on the data already published comparing erythrocyte and reticulocyte proteomes (Gautier et al. *Blood adv.* 2018), we explored other markers specific to reticulocytes (undetected in erythrocytes). We quantified Syntenin-1 (SCDBP) exclusively in CM samples (Dataset EV2). Additionally, Vimentin (VIM), another marker of reticulocyte, was also found to be more abundant in CM with a Log2foldchange of 3 compared to UM samples (Dataset EV5), but this observation did not reach statistical significance ($p=0.085$). These findings provide additional support to our hypothesis of increased reticulocytes in CM, initially based solely on CD71 (or TFRC).

Nevertheless, in the revised version of the manuscript we provide a targeted ELISA assay on 20 samples (10 UM vs 10 CM), in which we also observed an increased concentration of CD71 in CM infected erythrocytes whole cell extract from the NeuroCM cohort.

*2. Adding to the previous comment, did the authors observe higher number of uninfected reticulocytes in SM and/or CM samples? This was briefly mentioned in the discussion, but may need to be expanded. If there are infected and uninfected reticulocytes in a culture, the infected ones seem to mature faster than the uninfected ones according to recent literature (Ong et al, *Spectrochimica Acta* 2023). Hence, the argument that non-infected reticulocytes mature and infected ones remain the same as the authors suggest, are not in agreement.*

During the field study, we did not perform any blood cells count that included reticulocytes, since this investigation is not part of the conventional lab analysis performed in the collaborating hospitals. The question whether infected reticulocytes mature or not is a perspective that we wish to explore in further studies.

We clarified our hypothesis in the discussion (line 322-324), we suggest that while the reticulocytes mature, they maintain the *TFRC* receptor to sustain iron intake for *Plasmodium*.

We thank the reviewer for suggesting Ong et al.'s article, which shows a faster loss of CD71 in reticulocytes infected by *P. falciparum*, with complete disappearance of CD71 at the trophozoite stage (20 h.p.i). However, their results reflect an *in vitro* study based on reticulocyte culture and a laboratory strain of the 3D7 parasite. In our study, we analyze blood samples from patients, and the complex environment might lead to different outcomes in red cells. We modified the discussion (Line 421-423) to add the perspective from Ong's paper.

3. *Regarding accelerated growth of the parasites in CM condition as proposed by the authors, it is conceivable that the parasites mature faster due to suitable nutrient environment provided by host reticulocytes they are within. However, are they able to provide experimental evidence for this- such as stage-specific distribution of parasites at the beginning (mostly ring) and end of in vitro maturation step (18-32 hours later, when they should be trophozoites/schizonts), between the samples tested?*

No stage specific count was performed at the sample collection and post maturation, so we do not have data to support that hypothesis.

In another study published by our group using the same samples (from the NeuroCM cohort), we performed a transcriptomic study on circulating parasites and compared the age of the parasites according to malaria severity (Guillochon et al. JID 2022). In this study, we showed that circulating parasites in CM patients are in ring forms (9h.p.i) - approximately 90%, while UM patients have around 50% of parasites in young trophozoite forms (19h.p.i) in the samples. These previous published results show a shorter circulation time for the parasite in CM. These findings are in accordance with the literature and were previously described by Thomson-Luque et al. (Nat Commun 2021).

The hypothesis that the availability of more nutrients in reticulocytes promotes parasite maturation and could be related to the shorter circulating time is only a hypothesis that we mention in the discussion (line 333-337) but is not supported by our data.

4. *The manuscripts have several typos throughout, and needs to be corrected in a revision.*

We thank the reviewer for this suggestion, and we provide a revised version of the manuscript with thorough editing and proofreading.

Referee #2 (Comments on Novelty/Model System for Author):

My major concern refers to the analysis and conclusions derived from infected erythrocytes (iE). Student's t-tests ($p < 0.05$) without Benjamini-Hochberg correction were performed due to a lack of CM samples ($n = 3$) and lower quantification coverage. This raises doubts about the robustness of the differences between groups (high risk of false discovery rate). The conclusion about the preferential infection of reticulocytes or erythroid precursors by parasites causing CM, and the alteration of their maturation, is commented extensively in the discussion, without considering the potential limitation due to limited sample size. The wording used is very conclusive (ie, 648: "This preference enables the parasite to boost its growth, resulting in a faster increase of parasite load within the human body") and not sustained by the data. I recommend tuning down this part.

A similar issue of potential spurious results should be considered for the p values presented in Figure 4, which should be corrected for multiple comparisons between groups. Some of the observation (ie, $p=0.035$) have a high risk of being chance findings.

As these remarks are also mentioned later, we will comment them in the corresponding "Remarks for Author" section.

Referee #2 (Remarks for Author):

The manuscript by Fraering et al intitled "Infected erythrocytes and plasma proteomics reveals a specific protein signature of severe malaria" uses an unbiased proteomic assessment of infected erythrocytes (n=17) and plasma samples (n=24) collected from Beninese children, followed by targeted assays, to identify proteins differentially expressed between children with cerebral malaria (CM), severe anemia (SAM) and uncomplicated malaria (UM) as well as uninfected children. Up-regulation of the erythroid precursor marker transferrin receptor protein 1 (TFRC) was observed in CM infected erythrocytes. Increased levels of the 20S proteasome components were observed in plasmas from children with SM. Findings were confirmed using targeted assays in a cohort of 346 children. Authors conclude that the results suggest a) that parasites causing CM preferentially infect reticulocytes or erythroid precursors and alter their maturation, and b) that the host plasma proteome serves as a specific signature of SM which may be used to develop novel diagnostic and prognostic biomarkers.

The manuscript, which requires some grammar/syntaxes check, is very dense, repetitive and difficult to follow in several sections. The conclusions are very speculative. A higher synthesis, especially in the introduction and discussion, may increase the clarity of the paper. As stated in the abstract, the objective is highly ambitious ("to decipher the complex mechanisms underlying the pathophysiology of CM and the corresponding host-parasite interactions") and difficult to achieve with the limited number of samples included in the study.

We thank the reviewer for these comments. We improved the manuscript by removing the repetitions and corrected the grammar. We have also re worked the results section for more clarity. Briefly, we separated the LC-MS/MS analyses from the colorimetric analyses in the results section, removed a section discussing about LC-MS/MS targeted quantification (Figure 4B) that was adding unnecessary complexity to the message, and added a separate figure presenting all the colorimetric assays (Figure 5).

We agree with the reviewer that our discussion, primarily regarding the results derived from the analysis of iE, is speculative. These hypotheses arising from our results would need to be confirmed by *in vitro* experiments. This study was not designed to investigate the type of red blood cells the parasite infects during CM, and the observed CD71 abundance results in LC-MS/MS are an exciting but incidental finding. We reworded the discussion to emphasize that we are only providing hypotheses, and offer some suggestions to obtain experimental proofs of our assertions.

To bring more strength to our findings, we implemented a selective quantification of TFRC/CD71 by ELISA on $n = 20$ samples from the NeuroCM cohort. These results (showcased Line 210 to line 212 in the revised version) showed an increased TFRC concentration in CM (mean = 32 ng/mL) compared to UM (mean = 23 ng/mL). However, these results were not statistically significant ($p=0.19$) and a power calculation based on these results suggests to add 90 more samples (with a type II error β set to 0.15). Since these samples are really difficult to obtain from field studies, we unfortunately don't have enough samples to reach this threshold so we can only analyze tendencies.

Regarding the abstract, the reviewer is entirely correct, the wording of our sentence was too ambitious. We have revised it to: "**to study the complex...**" of the complex mechanisms underlying the pathophysiology of CM." (line 46-48).

My major concern refers to the analysis and conclusions derived from infected erythrocytes (iE). Student's t-tests ($p < 0.05$) without Benjamini-Hochberg correction were performed due to a lack of CM samples ($n = 3$) and lower quantification coverage. This raises doubts about the robustness of the differences between groups (high risk of false discovery rate).

We fully agree with the reviewer that the statistical analysis of iE bears a risk of false positive discovery due to the small sample size of the CM group. As a consequence, the dataset as it is cannot accommodate p -values corrections using B-H multiple testing since the variability remains relatively high within groups. This is especially limiting, for the CM containing only 3 samples. We thus chose to incorporate our iE results and conclusion based on p -value with a high risk of false positive rates.

These analyses were part of a discovery/screening phase analysis; in which it is accepted to use uncorrected p -values. This is exemplified by a recent and significant study in the malaria field, conducted by Mahamar et al. (Blood 2022), which explored a similar approach. This study also employed uncorrected p -values and relied on a Fold-change cut-off to identify differentially abundant proteins in a 3vs3vs3 comparison. In the revised manuscript, we provide a targeted quantification by ELISA of *TFRC* in 20 samples from the NeuroCM cohort in which we also observed an increased *TFRC* concentration in CM samples (described above) that also strengthens the LC-MS/MS results.

However, we agree with the reviewer that there is a risk of false discovery rate, and subsequently we updated the discussion to highlight that the small sample size in iE is a weakness (for both MS or the targeted *TFRC* ELISA) and that further investigations are needed (line 421-423). And we have clearly written (line 339-340) a sentence that aware readers about the sample size in infected erythrocytes.

The conclusion about the preferential infection of reticulocytes or erythroid precursors by parasites causing CM, and the alteration of their maturation, is commented extensively in the discussion, without considering the potential limitation due to limited sample size.

We agree with the reviewer and nuanced the discussion (no specific lines it's about phrasing in the whole discussion), and restated (line 339-340) that the small sample size is a limitation to the conclusion we raised.

The wording used is very conclusive (ie, 648: "This preference enables the parasite to boost its growth, resulting in a faster increase of parasite load within the human body") and not sustained by the data. I recommend tuning down this part.

We agree with the reviewer and modified the discussion accordingly adding conditional (line 419-420 and line 421-423).

A similar issue of potential spurious results should be considered for the p values presented in Figure 4, which should be corrected for multiple comparisons between groups. Some of the observation (ie, $p=0.035$) have a high risk of being chance findings.

We agree that a p -value of 0.035, especially without correction for multiple tests could be chance findings. This p -value correspond to the putative knob associated heat shock protein 40 -

PF3D7_0201800 for which no real speculative conclusion was addressed in discussion. We also added sentence (line 339-340) that clearly explains the risk of false discovery rate in iE.

Sometimes it is difficult to see if the results presented refer to the infected erythrocytes or plasma samples.

In the revised version of the manuscript, we reorganized the results to enhance clarity. We have also revised the original figures to better delineate the analyses. These adjustments in the results section should now improve clarity, allowing the reader to easily discern the biological compartment being examined.

Authors claim to have found a specific signature of severe malaria but they only included children with cerebral malaria and severe anemia. The use of severe malaria along the manuscript should be revised.

We thank the reviewer for this comment, and agree that this study do not cover all the various clinical manifestations of severe malaria (SM). Therefore, we have clarified at the beginning of the Materials and Methods: "In this study, we will use the term "SM" when analyzing the combination of CM and SMA samples "(Line 455-456). Of course, SM is a broader category and our study focuses solely on the two most common and severe forms (CM and SMA), also mentioned at the end of the introduction (line 129-131).

Line 308: "A total of 346 Beninese children were enrolled in this study". But this is not correct. Children were enrolled in a previous study, and samples were selected for this study. How was the selection done?

We modified the "Study location" section in Methods to emphasize that this study is the secondary analysis of a patient cohort (line 433-434).

For the LC-MS/MS analysis, the selection was based on technical factor, mainly the successful maturation and MACS enrichment, of the samples (specified in the manuscript line 151-152)

Regarding the targeted protein quantification confirmation analysis, all available samples (from all cohorts) were analyzed within the limits of remaining plasma samples available, which was our only selection criterion. For the non-infected samples, they were randomly selected from patients without fever and with negative TDR and PCR results.

We have added a sentence (line 140-141) in this results section to specify the sample numbers:

"Out of the 340 samples collected, 314 were available in sufficient quantity to conduct both LC-MS/MS and colorimetric analyses. No additional inclusion criteria were applied."

Grammar syntaxis: check "Samples preparation" (176),

Changed with: The samples were prepared for LC-MS/MS following the methodology previously described (31).

"this results" (631), Changed with: "These"

was (54, instead of were) and other errors in the document.

The manuscript has been reworded and revised for grammar and syntaxis errors. The context of this sentence changed.

Please also revise dots vs comma to express decimals.

Decimals are now correctly represented with dots.

Line 179: the iE resuspension (not only the proteins) were heated.

We have replaced the text with the wording suggested by the reviewer.

Clarity of some sections can be increased by removing repetitions. For example, the use of Student t-test and the $p < 0.05$ is mentioned two times in 257 and 278; the use of "65% of quantitative values in at least one group" is mentioned 4 times (234, 335, 341, 345); the identification of L-lactate dehydrogenase (pLDH -PF3D7_1324900) in all SM samples mentioned 2 times (627 and 474). All this increases the complexity of the document and could be minimized by a clear description in the methods.

We removed all repetitions listed by the reviewer, and overall improved the clarity of the manuscript.

397: Authors mention that they "confirmed the increase of FTL and the decrease of HPX in CM initially found by LC-MS/MS". However, this was not observed in the colorimetric results, as authors say. So, authors should use another word instead of confirmation to avoid driving the reader to confusion (may be observed).

This sentence has been rewritten and moved to an independent section (line 280-283) to improve clarity.

337: Authors mention that "the number of proteins in UM samples ($n=725$) was significantly lower compared with SMA ($n=1176$) and CM ($n=824$) ($p=0.007$; Kruskal-Wallis rank sum test)". May authors be referring to the median number of proteins per sample instead to the actual number of samples? Otherwise I do not understand how the statistical test was used.

The number within the parentheses represented the total count of identified proteins in each group, and the reviewer is right we analyzed the number of proteins per sample in each group. However, this section has been removed from the main part of the manuscript in the revised version, thus improving the clarity of the results section.

340: "For the purpose of conducting differential analysis, 619 proteins shared among all clinical presentations were selected by using a filter of 65% quantitative values in at least one group". I understand from this sentence that authors are excluding from the analysis those proteins which are not expressed on one or more groups. Can this lead to exclude highly differentially expressed proteins (ie, those that are not expressed in particular groups but may be expressed in others)?

We thank the reviewer for this comment and agree that our sentence "shared among all clinical presentation" was false and bring confusion to the comprehension of the filter, we removed this sentence.

The filter of 65% valid values in **at least one group** is a relatively common filter in the field of proteomics for selecting the most quantifiable proteins for differential analysis. This will allow a protein that got 4/6 values in the CM group for example to be kept even if this protein do not have values in the other groups.

Yes, this will lead to the exclusion of "**potential**" highly differentially abundant proteins. But since the absence of quantification values does not mean the absence of the protein in the sample, without quantification values these proteins could not be analyzed by statistical tests mandatory to establish a differential of expression. We added a dataset that regrouped all exclusive proteins we found.

474: Authors show that pLDH was detected in all infected samples. Later (632), author mention that "This results aligns with the diagnostics performances of pLDH in RDTs, which are less sensitive than RDTs detecting other proteins such as HRP2 (80)" Why HRP2 was not detected and differentially expressed, taking into account the very different parasite densities between groups?

To increase discussion clarity, we removed this part of the discussion.

To address the reviewer's question regarding HRP2, the protein was not detected by LC-MS/MS because the trypsin cleavage sites generated peptides that were either too large or too small (<6 amino acids) to be identified.

In addition, we removed the section regarding pLDH RDTs in the manuscript which was not contributing to the narrative.

The discussion is very long and difficult to identify clear key messages. Avoiding the repetition of results (551-553, 627 and others), a more synthetic presentation of ideas and a less speculative discussion may increase clarity.

We agree with the reviewer that the discussion was quite dense. We have removed several sections and significantly condensed others to emphasize the most important points. Additionally, we have carefully refined the key messages to make them more prominent in the discussion.

628: Something seems to be missing in this sentence: "Detection (79), which we identified in all SM plasma samples".

This sentence was an error on our part. We have removed it.

Referee #3 (Comments on Novelty/Model System for Author):

The authors used a well-established shotgun proteomics pipeline from sample preparation, data acquisition and analysis with an appropriate number of biological replicate analyses.

The medical impact stems from the potential identification of novel plasma biomarkers to better classify cases of severe malaria for earlier intervention, but how these protein signatures would be assayed in the field is not obvious.

We thank the reviewer for this suggestion and added in the discussion (line 423-428) a perspective regarding potential biomarkers confirmation and further use in the field. Nevertheless, our results, while exciting, require further confirmations before field testing.

Referee #3 (Remarks for Author):

To identify potential new biomarkers of severe malaria (SM), Fraering and colleagues report the proteomics analysis of proteins from infected erythrocytes (iE) and plasma obtained from clinical isolates of children under 5 diagnosed with cerebral malaria (CM) and severe malarial anemia (SMA) and compared with uncomplicated malaria (UM) and non-infected individuals (NI). The authors follow a well-established shotgun proteomics pipeline by digesting proteins into tryptic peptides, separating these offline (either using strong cation exchange for the iE samples or high pH reverse phase fractionation for the plasma samples), and analyzing the fractions by RPLC coupled to tandem mass spectrometry in data dependent acquisition mode. The MSMS datasets were searched against a combined Plasmodium falciparum and human protein database using MaxQuant and Perseus for protein identification and label-free quantitation. Validation studies

using the plasma of 2 large patient cohorts were performed via colorimetric assays to target and quantify proteins significantly shown to be dysregulated in SM in the LCMS datasets.

The authors chose to only report the proteins in plasma and iE that were assessed to be differentially abundant by pair-wise comparisons with controls (Supplemental Table S1). It would have been beneficial to also include a table with the entire list of identified and quantified proteins in each of the replicate analyses for the 3 clinical conditions and control. Only reporting p-values and log2 does not allow for evaluating reproducibility and variation between replicate analyses. In addition, reporting the complete list of proteins might help better understand the "65% quantitative values in at least one group" filtering criteria applied to "select proteins for downstream analysis". This filtering step is not clearly explained in the Materials & Methods section, e.g., this sentence on page 10, lines 234-235, could use reworking "proteins with less cross-replicates values were filtered out (65% of quantitative values in at least one group)".

These data have been deposited in the ProteomeXchange repository via PRIDE (Proteingroup file) and will be accessible to the scientific community. However, to further facilitate access to these data, we have added the raw output tables from the MaxQuant software as supplemental tables as suggested by the reviewer (Dataset EV1 and EV2).

Regarding the filters used, we have added a sentence in the methods section to clarify it (line 533-537). This filter is relatively commonly used in proteomic analysis and serves to reduce the number of simultaneous statistical tests for proteins that are poorly quantified (with many missing values) and for which the tests would, in any case, yield negative results due to high variations. We chose 65% so that 4 values in at least 1 group would be required to select a protein for plasma analysis.

Furthermore, the rationale for selecting proteins for differential analysis seems flawed since the authors only selected proteins "shared among all clinical presentations" (page 14, line 340). From the figure legend, the PCA analyses in Fig 2D-E were performed on 284 proteins quantified in all 4 conditions x 6 biological replicates of plasma samples. Since the CM condition had 114 unique proteins (Venn Diagram in Fig2A), I'd argue that including the proteins reproducibly quantified in the CM samples might have helped separate CM from SMA better.

We thank the reviewer for this comment and agree that our sentence "shared among all clinical presentation" was false and bring confusion to the comprehension of the filter, we removed this sentence.

We used a common filter for discovery data in LC-MS/MS: a minimum of 65% valid values in at least one of the four clinical groups was mandatory to select a protein. This means that a protein with 4 LFQ values in the NI group but not quantified in any other samples will be selected using this filter, but could not be analyzed through statistical tests. This type of selection is relatively common in proteomic **differential expression analyses** and does not appear to be flawed in our view. The only parameter that changes between studies is the percentage of valid values implemented, regarding our sample size in each group we chose 65%.

For the **Principal Component Analysis**, the aim of the study was to assess whether on the most easily quantifiable proteins, we could distinguish 'non-severe' cases from severe cases. To deal with missing values for certain proteins, we explored two common analytical approaches. The first approach, as we have done, is to filter by considering only the set of proteins with complete LFQ values (only measured by the machine) available for every sample (100% valid values). The second approach, less strict and similar to our method for protein differential abundance analysis, consists in selecting proteins with a minimum of XX% valid values in at least one group and imputing 'artificial' values (based on normal distribution) for the missing data. We also tried this method (which increased the number of proteins in the PCA to 781 - as in the differential

analysis), but it didn't change the sample clustering in PCA. Consequently, we decided to show the more stringent analysis based solely on values measured by the mass spectrometer.

The reviewer also mentioned the lack of exclusive proteins in this PCA: we totally agree with the reviewer that these proteins are important. However, we think that PCA is not the good place to implement these data that should be analyzed separately from quantitative data in a qualitative way only. As these proteins totally lack quantitative values in the other groups, we think that imputing a value of "0" or even imputing values based on normal distribution is something wrong regarding the biological reality.

Thus we added 2 distinct tables (Dataset EV2 and Dataset EV4) with exclusive proteins found in plasma and iE, that are available for the reader that we briefly overviewed in the manuscript (line 165-168 for iE and 170-172 for plasma).

Throughout the manuscript, several consistency issues with references to figures or supplemental tables not matching the sentences in the Results sections will need to be fixed/clarified:

Pg 16, In. 389-390: "we observed in CM plasma samples a significant ($q=0.017$) decrease of Hemopexin (HPX) abundance when compared to UM ($FC=-4.44$) and SMA ($FC=-2.96$) (Figure 4A)." While Hemopexin is indeed shown in Fig4A in the cluster of 21 proteins highly expressed in CM samples, hemopexin, along with all other proteins in this cluster, is not listed in the LC-MSMS supplemental table reporting differentially expressed proteins in plasma (Table S1B). The data for these proteins should be added to this supplemental table.

Here, we apologize because Table S1B had an active filter on one column. We have removed this filter, and now Hemopexin, along with other proteins previously obscured by the filter, is visible. We have also applied color filters to our supplemental tables to allow readers to examine the common differential proteins between groups and make their own filtering choices based on significantly dysregulated proteins. We also have added a column where Log2FC values have been transformed into Foldchange ($Foldchange = 2^{(\log_2 foldchange)}$) for each comparison to align with the text for all supplementary tables.

Pg 16, In. 391-393: "we found a significant increase of ferritin light chain (FTL) abundance in CM samples, when compared to UM ($p=0.0001$; $FC=7.50$) and SMA ($p=0,002$; $FC=5.58$) (Supplemental Table S1B)". Quantitative data for "Ferritin light chain" is not reported in S1B but rather is part of a set of 6 "inflammation biomarkers" listed in Table S1D. Why are these proteins not listed as differentially expressed in S1B is confusing.

We completely agree with the reviewer on this point. Initially, we considered extracting only known biomarkers, including *FTL*, and conducting statistical tests independently from the differential of expression on these biomarkers. However, this approach introduces statistical bias and does not appear relevant to include in the paper. Therefore, we have removed this section, including Table S1D, to significantly enhance clarity.

Pg. 16, In. 410-412: "analyzing the plasmatic proteome of malaria patients (six NI, six UM, six SMA, and six CM) and found 68 DAP ($q\ value < 0.05$) among the 781 selected proteins (Supplemental Table S1B)." Again, Table S1B, as provided, only contains 33 entries not 68.

Same response as for the previous comment regarding Hemopexin, Table S1B was filtered. We have removed this filter, and now we have a total of 68 proteins.

The authors used colorimetric assays to establish the concentration of 9 proteins in plasma

samples from their large cohorts (Table S2A, with boxplots for 5 proteins shown in Fig3C). While LC-MS/MS data is available for hemopexin (Fig4A) and ferritin light chain (TableS1D), it is unclear whether the other proteins tested by colorimetry were detected at all in the proteomics analyses since the authors only provide tables for filtered/differentially expressed proteins.

We have added a table (Dataset EV 3) containing all the plasma proteins found without undergoing any selection, we did the same for iE (Dataset EV1). Some proteins were specifically analyzed using colorimetry because we were unable to analyze them by LC-MS/MS (Depletion: Haptoglobin; Albumin and Transferrin). The proteins commonly quantified and analyzable are CRP, Hemopexin, Ferritin, and Hcpidin, although Hcpidin was poorly quantified using LC-MS/MS. We based the validation on these proteins, for which we observed a similar relative abundance between the LC-MS/MS and colorimetric data. But due to small sample in LC-MS/MS (6 per group) we could not do correlation analyses.

A side note: in Table S2A, colorimetric assays were performed on 355 plasma samples not 348 as written on line 395.

We checked and corrected the number 348 to 340 in line 137. We added the sentence, "Out of the 340 samples collected, 314 were available in sufficient quantity to conduct both LC-MS/MS and colorimetric analyses. No additional inclusion criteria were applied." (line 140-141). This sentence will enhance the precision regarding the sample selection used

Overall, while sample preparation, data acquisition and processing are technically sound, how the data is organized and discussed does not follow a logical flow. The panels assembled for Fig3 and 4 are particularly confusing: for example, data for hemopexin's concentration in plasma based on colorimetric assay is shown in Fig3C yet the proteomics data showing its high abundance in CM samples (leading to the subsequent colorimetric validation) is shown as a heat map in Fig4A. As a matter of course, the alpha-numerical labeling of figure panels should follow the order each panel is being discussed in the main text.

The reviewer is correct; we have significantly restructured the results section. Panels in Figures 3 and 4 have been modified to clearly separate the colorimetric from the LC-MS/MS analysis. We added a new figure (Figure 5) that exclusively includes the colorimetric analyses, with associated text moved to a new paragraph at the end of the results section (line 275). We hope to have improve the clarity of the result section. Regarding the discussion, we have made several modifications to enhance clarity.

In summary, I would strongly suggest the authors provide a supplemental table for the unfiltered quantitative protein lists for both the iE and plasma samples; tidy up their supplemental tables to match what is being stated in the main text; and reorganize results into sections that focus on 1 specific biological pathway with 1 figure containing panels with all relevant information for this group of proteins. This should help the readers make the most of this interesting study.

We thank the reviewer again for his contribution to the improvement of our manuscript. All suggestions were taken into account. We have added two tables that represent the raw protein quantification values from the MaxQuant analysis for both plasma (Dataset EV4) and iE (Dataset EV2) and the exclusive proteins of each group. The results section is now organized based on the LC-MS/MS analysis of iE and plasma then parasitic proteins and colorimetric validation. We hope to have brought more clarity with a better distinction between experiments and the studied compartment (plasma or iE).

9th Nov 2023

Dear Dr. Bertin,

Thank you for submitting your revised manuscript. We have now received the reports from the three referees who re-reviewed your manuscript, and as you will see below, they are supportive of publication. I will therefore be able to accept your manuscript once the following points will be addressed:

1/ Please address the minor comment from referee #1.

2/ Manuscript text:

- Please accept the previous changes and only keep in track changes any new modification.
- We would suggest removing "in this paper" in the first paragraph of the abstract.
- Materials and methods:
 - o Study approval: please add a statement that the experiments conformed to the principles set out in the WMA Declaration of Helsinki and the Department of Health and Human Services Belmont Report.
 - o Statistics: please include a statement about blinding, even if no blinding was done, and correct the checklist accordingly.
- Data Availability section: Thank you for providing reviewers' token. Please note that the datasets must be public before acceptance of the manuscript.
- Acknowledgements: Please merge the funding statement with the acknowledgements, under the heading "Acknowledgements".
- Please rename "Potential conflicts of interests" to "Disclosure statement and competing interests": We updated our journal's competing interests policy in January 2022 and request authors to consider both actual and perceived competing interests. Please review the policy <https://www.embopress.org/competing-interests> and update your competing interests if necessary.

2/ Figures:

- Please provide exact p values, not a range, in the figures or in their legends, including for ns - non-significant.
- Figure legends:

Please note that the error bars are not defined in the legend of figure 3b.
Please note that the measure of center for the error bars needs to be defined in the legend of figure EV1.

3/ Checklist:

- Statistics: please fill in the section on blinding.
- Ethics: please fill in the section on informed consent and Declaration of Helsinki.
- Data availability: if no publicly available data was re-used in the manuscript, but only in the response to reviewers, you do not need to fill this section.

4/ I slightly edited your synopsis text to match our style and format, please let me know if you agree with the following or amend as you see fit:

A mass spectrometry-based proteomic screening of both plasma and infected erythrocytes from Beninese children with various clinical manifestations of malaria (Uncomplicated malaria (UM), Severe Malarial Anemia (SMA), and Cerebral Malaria (CM)) was performed.

- An increase of TFRC abundance was observed in infected erythrocyte samples from CM patients when compared to UM patients, coupled with a decrease of plasmatic Transferrin levels.
- 20S proteasome abundance was increased in plasma from children with severe malaria (CM and SMA) compared to UM, suggests that it could serve as potential biomarker of severe malaria.
- A deregulated iron metabolism pathway and an increased hemolytic anemia were described in the severe malaria groups.
- Six *P. falciparum* proteins involved in red blood cell invasion were quantified specifically in plasma samples from severe malaria cases.

5/ As part of the EMBO Publications transparent editorial process initiative (see our Editorial at <http://embomolmed.embopress.org/content/2/9/329>), EMBO Molecular Medicine will publish online a Review Process File (RPF) to accompany accepted manuscripts.

This file will be published in conjunction with your paper and will include the anonymous referee reports, your point-by-point response and all pertinent correspondence relating to the manuscript. Let us know whether you agree with the publication of the RPF and as here.

I look forward to receiving your revised manuscript.

Yours sincerely,

Lise Roth

***** Reviewer's comments *****

Referee #1 (Comments on Novelty/Model System for Author):

In their revised manuscript, the authors have addressed the queries. They have also provided additional data to support their hypothesis. If the authors can address minor typos etc, the manuscript may be suggested for publication.

Referee #2 (Remarks for Author):

All the comments and questions I raised have been successfully addressed and do not have further comments.

Referee #3 (Comments on Novelty/Model System for Author):

While the medical impact of the plasma biomarker data reported in this study has the potential to high by guiding the design of a predictive tool to better classify severe cases of malaria, my previous rating of the medical impact as "medium" stands. As stated by the authors in the last sentence of their revised discussion, this work is foundational.

Referee #3 (Remarks for Author):

Fraering and colleagues have satisfactorily addressed this reviewer's questions and comments and edited their manuscript accordingly by addressing confusing figure panels, improving clarity, and providing complete datasets.

The authors addressed the minor editor editorial issues.

22nd Nov 2023

Dear Dr. Bertin,

Thank you for submitting your revised files. I am pleased to inform you that your manuscript is accepted for publication and is now being sent to our publisher to be included in the next available issue of EMBO Molecular Medicine!

If you have any questions, please do not hesitate to contact the Editorial Office.

Thank you for your contribution to EMBO Molecular Medicine, and congratulations on your interesting work!

Yours sincerely,

Lise Roth
